# Semi-supervised Knowledge Transfer Across Multi-omic Single-cell Data

**Fan Zhang[1], Tianyu Liu[2], Zihao Chen[3], Xiaojiang Peng[4],**
**Chong Chen[5], Xian-Sheng Hua[5], Xiao Luo[6,*], Hongyu Zhao[2]**
[1]Georgia Institute of Technology, [2]Yale University, [3]Peking University,
[4]Shenzhen Technology University, [5]Terminus Group, [6]University of California, Los Angeles
fanzhang@gatech.edu, tianyu.liu@yale.edu, g.e.challenger@pku.edu.cn
pengxiaojiang@sztu.edu.cn, chenchong.cz@gmail.com, huaxiansheng@gmail.com
xiaoluo@cs.ucla.edu, hongyu.zhao@yale.edu

## Abstract

Knowledge transfer between multi-omic single-cell data aims to effectively transfer cell types from scRNA-seq data to unannotated scATAC-seq data. Several approaches aim to reduce the heterogeneity of multi-omic data while maintaining the discriminability of cell types with extensive annotated data. However, in reality, the cost of collecting both a large amount of labeled scRNA-seq data and scATAC-seq data is expensive. Therefore, this paper explores a practical yet underexplored problem of knowledge transfer across multi-omic single-cell data under cell type scarcity. To address this problem, we propose a semi-supervised knowledge transfer framework named Dual label scArcity elimiNation with Cross-omic multi-samplE Mixup (DANCE). To overcome the label scarcity in scRNA-seq data, we generate pseudo-labels based on optimal transport and merge them into the labeled scRNA-seq data. Moreover, we adopt a divide-and-conquer strategy which divides the scATAC-seq data into source-like and target-specific data. For source-like samples, we employ consistency regularization with random perturbations while for target-specific samples, we select a few candidate labels and progressively eliminate incorrect cell types from the label set for additional supervision. Next, we generate virtual scRNA-seq samples with multi-sample Mixup based on the class-wise similarity to reduce cell heterogeneity. Extensive experiments on many benchmark datasets suggest the superiority of our DANCE over a series of state-of-the-art methods. Code is available at `https://github.com/zfkarl/DANCE`.

## 1 Introduction

In the realm of biology and medicine, many experimental methods [18, 25, 53, 51, 54, 63] using high-throughput sequencing technologies have emerged and characterize diverse properties of single cells. The predominant techniques are single-cell RNA-sequencing (scRNA-seq) [55, 88, 49] and single-cell ATAC-sequencing (scATAC-seq) [15, 75] for understanding complicated organisms and tissues at the single cell level. Specifically, scATAC-seq is an epigenomic profiling method designed to assess chromatin accessibility and provides an additional layer of information that complements scRNA-seq, enhancing the ability to comprehend epigenetic heterogeneity within complex tissues. However, identifying the cell types of scATAC-seq data is challenging because of the high sparsity, large dimensionality, and increasing scale of scATAC-seq data. Considering the lack of cell type annotations for scATAC-seq data, several methods [38, 36, 72] have been proposed to transfer cell

---

*Corresponding author.

types from fully-labeled scRNA-seq data to scATAC-seq data[2]. Among these methods, scJoint [38] introduces a scalable transfer learning framework that successfully incorporates atlas-scale scRNA-seq with scATAC-seq data based on kNN techniques. scBridge [36] integrates multi-omic data heterogeneously by identifying reliable scATAC-seq cells with smaller omic differences in comparison to scRNA-seq cells. scNCL [72] employs prior domain knowledge along with contrastive learning to address the challenge of heterogeneous features from different modalities.

Despite significant progress made by the aforementioned approaches, their success relies on a substantial amount of annotated scRNA-seq data. In reality, annotating both scRNA-seq data and scATAC-seq data is expensive and challenging. Therefore, this study explores a practical yet unexplored problem of knowledge transfer across multi-omic single-cell data under label scarcity in both scRNA-seq data and scATAC-seq data. In this study, only a small fraction of scRNA-seq data with cell types annotated, while a significant portion of both scRNA-seq data and scATAC-seq data lack annotations. In contrast to previous works assuming abundant labeled scRNA-seq data, this setting aligns more closely with practical scenarios.

Addressing this realistic problem requires the consideration of two crucial aspects. Firstly, *how to learn representations with distinct cell type discriminability under label scarcity in both scRNA-seq data and scATAC-seq data?* Learning representations with good discriminability relies on a large amount of annotated data, yet collecting labeled data is extremely costly and difficult. Therefore, it poses a significant challenge in situations where both scRNA-seq data and scATAC-seq data lack a substantial amount of labels. Secondly, *how to reduce cell heterogeneity between different omic data while maintaining cell type discriminability?* Another challenge is how to reduce the heterogeneous gap between different omic data, which is crucial for cell type transfer. Additionally, while reducing cell heterogeneity, there is a high likelihood of compromising cell type discriminability. Even worse, this becomes significantly more challenging under label scarcity.

To answer these questions, we propose a semi-supervised knowledge transfer framework termed Dual label scArcity elimiNation with Cross-omic Multi-samplE Mixup (`DANCE`). Since the class distribution of the whole scRNA-seq data is unattainable, we start by injecting semantic knowledge using labeled source scRNA-seq data, and then generate pseudo-labels based on optimal transport (OT) for unlabeled scRNA-seq data. Afterward, the unlabeled scRNA-seq data, along with pseudo-labels, are incorporated into the labeled set. For unlabeled target scATAC-seq data, due to the heterogeneous gap, we adopt a divide-and-conquer strategy, which separates the whole data into source-like data and target-specific data. For source-like samples, we employ consistency regularization to force the model to make consistent predictions after random perturbations. For target-specific samples, we select ambiguous labels and then filter the incorrect labels, which can guide the optimization in a soft manner. To mitigate the heterogeneous gap, we propose to generate virtual scRNA-seq samples by multi-sample Mixup according to class-wise similarity and then we perform a sample-to-sample alignment strategy. The effectiveness of `DANCE` is validated by adopting a series of benchmark datasets and conducting comprehensive experiments in comparison to various state-of-the-art approaches.

To summarize, the main contribution of this paper includes: (1) *New Problem.* We study a realistic yet underexplored problem of knowledge transfer across multi-omic single-cell data under label scarcity in both scRNA-seq and scATAC-seq data. This problem holds significant implications for reducing the annotation cost of single-cell data. (2) *Novel Methodology.* We propose a semi-supervised framework named `DANCE` for this problem. `DANCE` introduces OT-based dataset expansion and a divide-and-conquer strategy for dual label scarcity elimination. Additionally, `DANCE` employs an effective alignment strategy of cross-omic multi-sample Mixup to reduce cell heterogeneity. (3) *Comprehensive Experiments.* Through extensive experiments with different settings on various benchmark datasets, we demonstrate the superiority of `DANCE` against many state-of-the-art methods.

## 2 Related Work

### 2.1 Multi-omic single-cell Data Integration

The integration of multi-omic single-cell data constitutes a fundamental challenge in elucidating biological processes. Multi-omic single-cell data affords a multitude of perspectives on cellular functions, thereby augmenting our understanding of biology. Some approaches [69, 44, 59, 37,

---

[2]We use scATAC-seq as an example and there could be other biological layers.

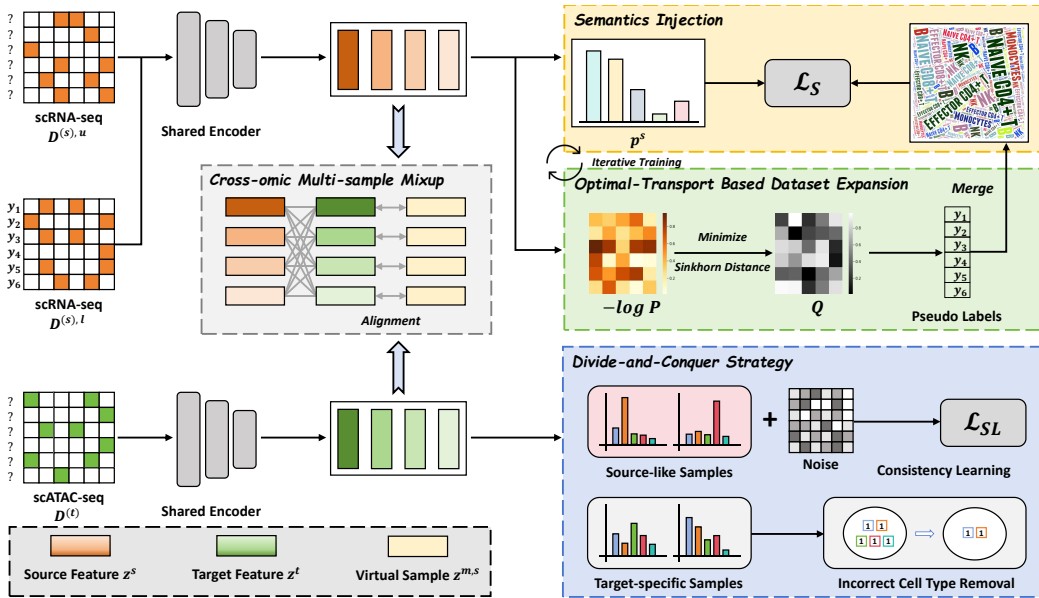

Figure 1: An overview of DANCE. We first utilize labeled scRNA-seq samples to inject semantics and expand the scRNA-seq dataset by OT-based pseudo-labeling. Then we employ consistency regularization and ambiguous set learning for source-like and target-specific scATAC-seq samples. In addition, we perform cross-omic multi-sample Mixup to reduce cell heterogeneity.

30, 61, 2] opt for denoising, batch correction, and integrating single-cell data across numerous experiments, encompassing transcriptomic data and scATAC-seq data. Nevertheless, the simple application of these methodologies to multi-omic data integration poses computational challenges and frequently yields unsatisfactory results, given the considerable differences in dimensions and sparsity levels among different modalities. Moving a step closer, methodologies such as scAI [25] and MOFA+ [3] utilize factor analysis and joint clustering to concurrently measure multi-omic data within the same cell. Nevertheless, these methods require pairwise data format, and executing paired measurements presents technical challenges and involves high costs. Therefore, another series of approaches have been proposed to measure multi-omic data from different cells, including manifold alignment [63, 1, 39], matrix factorization [18, 64, 84], correlation-based methods [5, 54], and neural network approaches [38, 36, 72]. Recent approaches often map multi-omic data into low-dimensional spaces through shared auto-encoders [21, 66, 56, 20], and Graph Neural Networks (GNNs) encoders [65, 60, 40], which focus on knowledge transfer across multi-omic data. For instance, scBridge [36] maps multi-omic single-cell data into a shared embedding space and mines reliable samples for dataset expansion, thereby merging the scATAC-seq and scRNA-seq data into the same dataset. GLUE [8] combines a knowledge-based graph and adversarial alignment to explore regulatory interaction across various omic layers. scCLIP [68] introduces a contrastive learning approach to integrate multi-omic single-cell data. It aligns the representations of pairwise multi-omic single-cell data without the usage of cell type labels. Despite their significant strides in integrating heterogeneous cells, all these methods overlook a practical challenge in reality, i.e., the difficulty of annotating both scRNA-seq data and scATAC-seq data. Therefore, there is an urgent need for an approach that can effectively alleviate label scarcity in knowledge transfer across multi-omic single-cell data.

## 2.2 Semi-supervised Learning

To tackle the label scarcity issue in reality, semi-supervised learning has been widely studied with various applications, such as image classification [46, 23, 7, 6], semantic segmentation [48, 14], and object detection [24, 80]. Recent studies on semi-supervised learning include consistency regularization approaches [46, 52, 67] and pseudo-labeling approaches [6, 7, 23, 79]. Consistency regularization usually first introduces random perturbations from various sources, including data input [67], network parameters [47], and deep features [28]. By encouraging the model to make consistent predictions under these perturbations, it can still learn good representations even under resource constraints. Pseudo-labeling typically assigns pseudo-labels to unlabeled samples using a

neural network. It often comes with some special techniques, such as dynamic thresholding [78, 33] and curriculum learning [29, 22, 10]. Then those pseudo-labels together with their corresponding samples are added to the dataset for future use. Besides the advanced approaches in computer vision, other related areas such as node classification [13, 62] and reinforcement learning [87] also benefit from semi-supervised learning. However, to the best of our knowledge, there exists no previous work to study the problem of semi-supervised knowledge transfer across multi-omic single-cell data. Therefore, we provide a semi-supervised framework termed DANCE to promote the understanding of semi-supervised learning in the field of bioinformatics.

## 3 Problem Definition

In our problem, we have two datasets $\mathcal{D}^{(s)}$ and $\mathcal{D}^{(t)}$, which denote the source scRNA-seq dataset and the target scATAC-seq dataset, respectively. In the semi-supervised setting, the scRNA-seq dataset $\mathcal{D}^{(s)}$ is separated into two parts, the labeled dataset $\mathcal{D}^{(s),l}$ and the unlabeled dataset $\mathcal{D}^{(s),u}$, where $\mathcal{D}^{(s),l} = \{(\boldsymbol{x}_i^{(s),l}, y_i^{(s),l})\}_{i=1}^{N^l}$ consists of $N^l$ scRNA-seq samples $\boldsymbol{x}_i^{(s),l}$ and the corresponding labels $y_i^{(s),l}$, and $\mathcal{D}^{(s),u} = \{(\boldsymbol{x}_i^{(s),u})\}_{i=1}^{N^u}$ includes $N^u$ unlabeled scRNA-seq samples $\boldsymbol{x}_i^{(s),u}$. The target scATAC-seq dataset with the size $N^t$ is denoted as $\mathcal{D}^{(t)} = \{(\boldsymbol{x}_i^{(t)})\}_{i=1}^{N^t}$, which includes $N^t$ scATAC-seq samples $\boldsymbol{x}_i^{(t)}$. We focus on the closed-set setting, thus both scRNA-seq data and scATAC-seq data are assumed to share the same set of $C$ cell types. A shared encoder $F(\cdot)$ is adopted to map both the scRNA-seq samples and scATAC-seq samples into the embedding space. Following the encoder, we utilize a classifier $H(\cdot)$ to convert representations into softmax predictions corresponding to different cell types. The representations and predictions are denoted as $\boldsymbol{z}_i = F(\boldsymbol{x}_i)$ and $\boldsymbol{p}_i = softmax(H(\boldsymbol{z}_i))$, respectively. It should be noted that only a small portion of the scRNA-seq data is labeled, and a significant number of both scRNA-seq samples and scATAC-seq samples lack cell type annotations. We aim to transfer the semantic information related to different cell types from $\mathcal{D}^{(s)}$ to $\mathcal{D}^{(t)}$, which has two expectations: *(1) The cell representations of scRNA-seq data and scATAC-seq data are mapped into a shared common embedding space with the similarity structure preserved. (2) The cell type annotations on scATAC-seq data are accurate after label transfer.*

## 4 The Proposed Approach

### 4.1 Framework Overview

This paper investigates the problem of semi-supervised knowledge transfer across multi-omic single-cell data. We employ a shared encoder and classifier for heterogeneous scRNA-seq data and scATAC-seq data. The labeled scRNA-seq data are utilized to inject the semantic knowledge. To overcome label scarcity in source scRNA-seq data, we generate pseudo labels based on optimal transport and merge them into the labeled set. To address label scarcity in target scATAC-seq data, we perform consistency regularization for source-like samples and learn from ambiguous labels for target-specific samples. To reduce the cell heterogeneity, we generate virtual scRNA-seq samples through cross-omic multi-sample Mixup and align the heterogeneous multi-omic data. The overview of the proposed framework is illustrated in Figure 1, and each component will be elaborated in the next sections.

### 4.2 Optimal Transport-based Dataset Expansion for Source scRNA-seq Data

The first obstacle is the label scarcity in scRNA-seq data. Recently, pseudo-labeling is a commonly used technique in semi-supervised learning [52, 78] and domain adaptation [83, 31] problems, which usually operate at the sample level [52, 83] by assigning the label with the highest confidence to each sample. However, this sample-level prediction-based pseudo-labeling strategy could be biased to easy classes, leading to biased and suboptimal results. To tackle this, we incorporate optimal transport (OT) [12, 9, 11] into the pseudo-labeling process, which is capable of taking into account prior cell type distributions from a global view to reduce potential noise in pseudo-labeling.

Here we first recall the knowledge of OT. OT is a constrained optimization problem designed to discover the optimal coupling matrix $\boldsymbol{Q}$ for mapping one probability distribution to another, with the goal of minimizing the overall cost. To represent the cost of transporting data from $\boldsymbol{\gamma}$ and $\boldsymbol{\eta}$, we introduce a cost matrix $\boldsymbol{C} \in \mathbb{R}^{|\boldsymbol{\gamma}| \times |\boldsymbol{\eta}|}$, where $|\boldsymbol{\gamma}|$ and $|\boldsymbol{\eta}|$ denote the dimension of $\boldsymbol{\gamma}$ and $\boldsymbol{\eta}$,

respectively. By applying the Sinkhorn algorithm [17] to minimize the Sinkhorn distance, we can obtain the optimal coupling matrix $\boldsymbol{Q}$:

$$\min_{\boldsymbol{Q} \in \prod(\boldsymbol{\gamma}, \boldsymbol{\eta})} \sum \boldsymbol{C}_{ij}\boldsymbol{Q}_{ij} + \frac{1}{\sigma}(-\boldsymbol{Q}_{ij}log\boldsymbol{Q}_{ij}), \tag{1}$$

where $\prod(\boldsymbol{\gamma}, \boldsymbol{\eta}) = \left\{ \mathbf{Q} \in \mathbb{R}_+^{|\gamma| \times |\eta|} \mid \mathbf{Q}\mathbf{1}_{|\eta|} = \boldsymbol{\gamma}, \mathbf{Q}^\top \mathbf{1}_{|\gamma|} = \boldsymbol{\eta} \right\}$ and $\sigma$ is set to 10 empirically.

To employ OT for pseudo-labeling, we begin by training the encoder and classifier on labeled scRNA-seq data, enabling the injection of semantic knowledge into the model. Similar to traditional cell type annotation problems [16, 77], we leverage a small number of labeled scRNA-seq data and minimize the cross-entropy between the classifier's output and the cell type labels:

$$\mathcal{L}_S = \sum_{\boldsymbol{x}_i^{(s),l} \in \mathcal{D}^{(s),l}} \text{CE}(\boldsymbol{p}_i^{(s),l}, y_i^{(s),l}), \tag{2}$$

where $\boldsymbol{p}_i^{(s),l}$ represents the predicted distribution of $\boldsymbol{x}_i^{(s),l}$.

Then, we aim to transfer the semantic information from labeled scRNA-seq samples to unlabeled ones. In particular, a mini-batch of $B$ unlabeled samples is fed to our neural network to generate the prediction matrix $\boldsymbol{P} \in \mathbb{R}^{B \times C}$, which could be biased to easy classes and thus inaccurate. To tackle this, we learn an optimal coupling matrix that should obey prior cell type distributions from a global view. Since we do not have prior information, we adopt a uniform distribution as in the polytope $\prod(\frac{1}{B}\mathbf{1}_B, \frac{1}{C}\mathbf{1}_C)$, where $\mathbf{1}_B$ and $\mathbf{1}_C$ indicate two vector of ones with dimensions of $B$ and $C$, respectively. In addition, the optimal $\boldsymbol{Q}$ should be close to $\boldsymbol{P}$. Therefore, we define the cost matrix $\boldsymbol{C}$ as $-log\boldsymbol{P}$, and Eqn. 1 can be reformulated as:

$$\min_{\boldsymbol{Q} \in \prod(\frac{1}{B}\mathbf{1}_B, \frac{1}{C}\mathbf{1}_C)} \sum_{ij} -log\boldsymbol{P}_{ij}\boldsymbol{Q}_{ij} + \frac{1}{\sigma}(-\boldsymbol{Q}_{ij}log\boldsymbol{Q}_{ij}). \tag{3}$$

The coupling matrix $\boldsymbol{Q}$ obtained by optimizing Eqn. 3 can be used to generate pseudo-labels for unlabeled scRNA-seq data. Formally, the pseudo-labels can be defined as:

$$\hat{y}_i^{(s),u} = argmax_j \boldsymbol{Q}_{ij}. \tag{4}$$

Then the unlabeled scRNA-seq samples and their corresponding pseudo-labels would be added to the labeled scRNA-seq dataset:

$$\mathcal{D}^{(s),l} \leftarrow \cup_{i=1}^{N^u}\{(\boldsymbol{x}_i^{(s),u}, \hat{y}_i^{(s),u})\} \cup \mathcal{D}^{(s),l}. \tag{5}$$

Leveraging OT-based pseudo-labeling for dataset expansion can effectively mitigate label scarcity in scRNA-seq data. On the one hand, we employ labeled scRNA-seq data for semantic injection using Eqn. 2, obtaining high-quality pseudo-labels. On the other hand, with the dataset expanded during the current iteration, more information is available for Eqn. 2 in the subsequent iteration. This leads to generating more reliable pseudo-labels in the following iterations. Through continuous iterative training, we can progressively enhance both the scale and quality of the scRNA-seq dataset. Next, we theoretically support the superiority of OT-based dataset expansion. See proof in Appendix A.

**Theorem 4.1.** *Let $\boldsymbol{P}_n = \frac{1}{B}\boldsymbol{P}$ be the normalized version of $\boldsymbol{P}$. Then after our optimal transport-based dataset expansion, we have*

*(1) For any $\varepsilon > 0$ and large enough $\sigma$, we have $\text{CE}(\boldsymbol{P}_n, \boldsymbol{P}) + \varepsilon \geq \text{CE}(\boldsymbol{Q}, \boldsymbol{P}) + C_{\boldsymbol{P}}$, where $C_{\boldsymbol{P}} = \sum_{j=1}^{C}\left(\frac{1}{B}\sum_{i=1}^{B}p_{ij} - \frac{1}{C}\right) \cdot \left(\frac{1}{B}\sum_{i=1}^{B} - \log p_{ij}\right)$ and $\text{CE}(\boldsymbol{Q}, \boldsymbol{P}) = \sum_{i,j}\text{CE}(\boldsymbol{Q}_{ij}, \boldsymbol{P}_{ij})$.*

*(2) Let $\boldsymbol{L} = (l_{ij})_{B \times C}$ denote the soft-version of true cell labels, where $l_{ij} = \max\left\{\mathbf{1}_{\{cell\ i\ belongs\ to\ class\ j\}}, \delta\right\}$ and $\delta > 0$. Suppose $0 < C_{\min} \leq l_{ij}/\boldsymbol{P}_{ij} \leq C_{\max}$ for all $1 \leq i \leq B$ and $1 \leq j \leq C$. Then, $\text{CE}(\boldsymbol{P}_n, \boldsymbol{L}) + \varepsilon + C_{\boldsymbol{L}} \geq \text{CE}(\boldsymbol{Q}, \boldsymbol{L}) + C_{\boldsymbol{P}}$, where $C_{\boldsymbol{L}} = \log C_{\max} - \log C_{\min}$.*

**Remarks.** $C_{\boldsymbol{P}}$ characterizes how far $\boldsymbol{P}_n$ is from $\prod(\frac{1}{B}\mathbf{1}_B, \frac{1}{C}\mathbf{1}_C)$. If the mini-batch was uniformly sampled across cell-types, then $\frac{1}{B}\sum_{i=1}^{B}p_{ij}$ should be close to $\frac{1}{C}$. Thus, $C_{\boldsymbol{P}}$ will be close to zero. Moreover, for those hard classes, $p_{ij}$'s are close to $\frac{1}{C}$, hence both $\frac{1}{B}\sum_{i=1}^{B}p_{ij}$ and $\sum_{i=1}^{B} - \log p_{ij}$ will not be too large. Hence, on the whole, $C_{\boldsymbol{P}}$ can be relatively small, which demonstrates that our optimization has the potential to achieve the desired distribution.

### 4.3 Divide and Conquer for Target scATAC-seq Data

Through OT-based pseudo-labeling, we effectively address the label scarcity issue in scRNA-seq data. However, due to the heterogeneous gap, it would not be optimal when employing a similar approach for scATAC-seq data. Here, to tackle label scarcity in scATAC-seq data, we adopt a divide-and-conquer learning strategy, which first divides the scATAC-seq data into source-like data and target-specific data and then utilizes separate approaches for optimization.

In particular, by comparing the confidence of the predictions and a threshold, we divide scATAC-seq data into two parts $\mathcal{D}^{SL}$ and $\mathcal{D}^{TS}$:

$$\mathcal{D}^{SL} = \{\boldsymbol{x}_i \in \mathcal{D}^{(t)}| \max_c \boldsymbol{p}_i^{(t)}[c] > \tau\}, \mathcal{D}^{TS} = \mathcal{D}^{(t)}/\mathcal{S}^{SL}, \tag{6}$$

where $\boldsymbol{p}_i^{(t)}$ is the predicted distribution of $\boldsymbol{x}_i^{(t)}$. The threshold $\tau$ is used to control the size of two groups and set to 0.9 according to previous works [52, 36]. These source-like samples are more likely to have accurate pseudo-labels with high confidence using the pre-trained model. In contrast, target-specific data are far away from source data distribution and thus the predictions could be noisy. Therefore, we utilize separate strategies for these two parts.

**Learning from Source-like Data.** For these source-like samples, we utilize consistency regularization to provide supervision [4, 32, 50, 52, 85]. Here, we add random perturbations (e.g., Gaussian noise) to data, and then force the network to make consistent predictions for both original and perturbed scATAC-seq samples. In particular, we can use $\hat{y}_i = argmax(\boldsymbol{p}_i)$ to obtain the pseudo-label of the original scATAC-seq sample. After that, we can get the prediction $\tilde{\boldsymbol{p}}_i$ of the perturbed scATAC-seq sample. Thus, the supervised loss can be formulated as:

$$\mathcal{L}_{SL} = -\frac{1}{|\mathcal{D}^{SL}|} \sum_{i=1}^{|\mathcal{D}^{SL}|} \sum_{c=1}^{C} \hat{y}_i^{\ c} log(\tilde{\boldsymbol{p}}_i^c), \tag{7}$$

where $|\mathcal{D}^{SL}|$ represents the number of unlabeled scATAC-seq samples in $\mathcal{D}^{SL}$. In this way, we combine pseudo-labeling with consistency learning to learn from source-like scATAC-seq samples.

**Learning from Target-specific scATAC-seq Data.** However, due to cell heterogeneity between scRNA-seq and scATAC-seq data, many scATAC-seq samples would be far away from the source modality. They are difficult to predict with low confidence scores [71, 85]. To tackle this challenge, we introduce a soft way to learn from their ambiguous label sets instead of pseudo-labels. Here, for each target-specific scATAC-seq sample, we first select the top-$k$ classes with the highest probabilities to make up the ambiguous label set and then introduce a soft supervised loss for reliable guidance.

To be specific, we first recall the predicted distribution $\boldsymbol{p}_i$ for each sample $\boldsymbol{x}_i$ in $\mathcal{D}^{TS}$. The negative cell types are removed with a significant difference from the cell type with the highest probability. In formulation, the removed labels should satisfy the following conditions:

$$\max_c \boldsymbol{p}_i[c] - \boldsymbol{p}_i[j] \geq \mu, \tag{8}$$

where the threshold $\mu$ is initially set to $1e-3$ and gradually decreases during the training process. The left labels make up the candidate label set:

$$Y_i = \{c \in Topk(\boldsymbol{p}_i)| \max_c \boldsymbol{p}_i[c] - \boldsymbol{p}_i[j] < \mu\}. \tag{9}$$

Then, we employ a soft supervision loss to provide reliable guidance for target-specific samples:

$$\mathcal{L}_{TS} = -\frac{1}{\mathcal{D}^{TS}} \sum_{i=1}^{\mathcal{D}^{TS}} \sum_{j=1}^{k} \alpha_{ij} 1_{j \in Y_i} log(\boldsymbol{p}_i[j]), \tag{10}$$

$$\alpha_{ij} = \begin{cases} \boldsymbol{p}_i[j]/\sum_{j \in Y_i} \boldsymbol{p}_i[j], & j \in Y_i, \\ 0, & otherwise. \end{cases} \tag{11}$$

where $|\mathcal{D}^{TS}|$ denotes the number of target-specific samples. The weight $\alpha_{ij}$ can allow the model to pay more attention to potential correct cell types with high probabilities.

Next, we provide the theoretical underpinnings to show that the quality of the soft labels from the candidate label set is higher than that of hard labels. The proof can be found in Appendix A.

**Theorem 4.2.** *Suppose we randomly draw $n$ samples $(X_1, \ldots, X_n)$ from two classes $\{0, 1\}$, where the proportion of class 1 is $p$. Further, suppose we can only observe a flawed version of these class labels $\boldsymbol{Z} = (Z_1, \ldots, Z_n)$ with error rate $\beta > 0$. That is, $\mathbb{P}(Z_i \neq X_i) = \beta$. Under loss $\mathcal{L}(p, \boldsymbol{z}) = \sum_{i=1}^{n} z_i \log p + (n - \sum_{i=1}^{n} z_i) \log(1 - p)$, consider two estimates of $p$:*

*(1) $p_{\mathrm{hard}} = \underset{p}{Argmin} \ \mathcal{L}(p, \boldsymbol{Z})$.*

*(2) $p_{\mathrm{soft}}^t = \underset{p}{Argmin} \ \mathcal{L}(p, t\boldsymbol{Z} + (1 - t)(1 - \boldsymbol{Z}))$, where $t \in (0, 1)$.*

*Then, there exists $t_0 \in (0, 1)$ such that for all $t \in (t_0, 1)$, we have $\mathbb{E} \left\| p_{\mathrm{soft}}^t - p \right\|^2 \leq \mathbb{E} \left\| p_{\mathrm{hard}} - p \right\|^2$.*

**Remarks.** In Eqn. 10, if we take $\boldsymbol{p}_i[1] = p, \boldsymbol{p}_i[0] = 1 - p$, and $\alpha_{ij} = t$ if $j = Z_i$, then the estimation under Eqn. 10 will degenerate to $p_{\mathrm{soft}}^t$.

### 4.4 Cross-omic Multi-sample Mixup for Cell Heterogeneity Reduction

Despite the proposed techniques in addressing the label scarcity problem, reducing the heterogeneous gap between scRNA-seq data and scATAC-seq data remains an additional challenge. To tackle this challenge, previous works [27, 19, 35, 34, 70] propose many strategies for cross-modal alignment. However, these strategies have two major prerequisites. On the one hand, strategies based on centroid learning [27, 19] require accurate label information from both source and target modalities to aggregate samples of the same class. On the other hand, strategies based on cross-modal contrastive learning [35, 34, 82] require pairwise information. Neither of these prerequisites is applicable to our problem, as pairwise information is difficult to acquire in single-cell data, and the majority of annotation information is missing. Towards this end, we propose a sample-to-sample alignment strategy termed cross-omic multi-sample Mixup [81, 58, 86, 73], which first calculates the class-wise similarity and then mixes the scRNA-seq samples in the hidden space for cross-omic alignment.

In particular, we transform the intrinsic distribution of the source scRNA-seq samples to align with scATAC-seq samples in the hidden space. Since we cannot get accurate pairwise information, we calculate the similarity between scRNA-seq samples and scATAC-seq samples and generate virtual samples instead. Note that the weights of the classifier are considered as prototypes that contain representative information of different cell types. We first form a weight matrix $\mathcal{W} = [\boldsymbol{w}_1, \boldsymbol{w}_2, \cdots, \boldsymbol{w}_C]^T \in \mathbb{R}^{C \times D}$, where $D$ denotes the hidden dimension of the embeddings. Then, the class-wise similarity can be determined using cosine distance:

$$S_{ij} = cos(\boldsymbol{p}_i^T \mathcal{W}, \boldsymbol{p}_j^T \mathcal{W}) = |\boldsymbol{p}_i^T \mathcal{W}| \star |\boldsymbol{p}_j^T \mathcal{W}|^T, \tag{12}$$

where $|\cdot|$ represents the $L_2$ normalization. For a batch of $B^{(s)}$ scRNA-seq samples and a batch of $B^{(t)}$ scATAC-seq samples, we can obtain a similarity matrix $S^B \in \mathbb{R}^{B^{(s)} \times B^{(t)}}$ using Eqn. 12. For the sake of simplicity, we let $B^{(s)} = B^{(t)} = B$, and the similarity matrix within a mini-batch can be formulated as $S^B \in \mathbb{R}^{B \times B}$. Afterwards, we employ the softmax function to convert $\mathcal{S}$ into the weight matrix $\mathcal{M}$ for Mixup, which corresponds to the probability of sharing the same semantics:

$$\mathcal{M} = softmax(\mathcal{S}), \tag{13}$$

where $\mathcal{M}_i$ collects the weights of $B$ scATAC-seq samples corresponding to the $i$-th scRNA-seq sample. Then, we can fuse the scRNA-seq samples into a virtual sample through Mixup:

$$\boldsymbol{z}_i^{m,s} = \sum_{j=1}^{B} \mathcal{M}_{ij} \cdot \boldsymbol{z}_j^s. \tag{14}$$

After getting the fused scRNA-seq samples, we can perform a sample-to-sample alignment strategy. The loss function can be formulated as:

$$\mathcal{L}_A = \frac{1}{B} \sum_{i=1}^{B} ||\boldsymbol{z}_i^{m,s} - \boldsymbol{z}_i^t||^2. \tag{15}$$

Table 1: Quantitative comparisons (in %) with state-of-the-art approaches with various label ratios. **Bold** numbers indicate beset results, and underlined numbers indicate second-beset results.

| Label Ratio | Low | Mid | High | Low | Mid | High | Low | Mid | High | Low | Mid | High |
|---|---|---|---|---|---|---|---|---|---|---|---|---|
| Data | snRNA_10X_v3_A-ATAC | | | snRNA_10X_v3_A-snmC | | | snRNA_10X_v2-ATAC | | | snRNA_10X_v2-snmC | | |
| DAN [42] | 20.56 | 20.13 | 20.33 | 44.20 | 46.55 | 53.72 | 27.50 | 29.14 | 30.47 | 54.33 | 55.08 | 49.95 |
| CDAN [43] | 23.46 | 21.29 | 21.84 | 20.38 | 26.00 | 27.76 | 27.67 | 22.71 | 21.51 | 39.08 | 32.97 | 38.08 |
| MCC [26] | 24.34 | 32.39 | 40.84 | 20.40 | 48.69 | 44.59 | 24.57 | 31.45 | 33.03 | 31.22 | 36.77 | 40.20 |
| FixMatch [52] | 24.38 | 33.45 | 41.60 | 20.42 | 44.58 | 41.98 | 24.93 | 31.61 | 33.04 | 31.66 | 36.60 | 41.32 |
| scJoint [38] | 17.58 | 19.52 | 18.86 | 23.46 | 24.27 | 21.86 | 19.39 | 18.51 | 17.78 | 21.48 | 24.43 | 22.11 |
| scNCL [72] | 22.62 | 21.21 | 22.92 | 27.16 | 59.78 | 60.07 | 21.98 | 28.11 | 30.19 | 59.65 | 67.47 | 68.89 |
| scBridge [36] | 25.57 | 38.50 | 41.86 | 21.64 | 32.74 | 35.71 | 19.39 | 16.29 | 42.36 | 12.70 | 20.85 | 38.48 |
| Ours | **50.23** | **51.02** | **51.42** | **75.72** | **78.67** | **80.12** | **50.83** | **52.90** | **54.79** | **60.12** | **72.24** | **78.04** |
| Data | snRNA_SMARTer-ATAC | | | snRNA_SMARTer-snmC | | | scRNA_SMARTer-ATAC | | | scRNA_SMARTer-snmC | | |
| DAN [42] | 22.02 | 27.07 | 28.06 | 48.28 | 55.93 | 57.67 | 21.23 | 20.02 | 20.31 | 41.96 | 54.31 | 52.39 |
| CDAN [43] | 20.99 | 28.65 | 30.07 | 33.87 | 35.37 | 51.19 | 22.05 | 21.50 | 21.58 | 34.66 | 42.43 | 38.70 |
| MCC [26] | 20.99 | 23.41 | 24.09 | 34.37 | 52.46 | 55.38 | 21.73 | 21.30 | 21.54 | 32.09 | 50.35 | 49.24 |
| FixMatch [52] | 21.25 | 24.23 | 24.88 | 33.80 | 52.05 | 54.78 | 22.07 | 21.77 | 22.32 | 31.55 | 49.98 | 49.38 |
| scJoint [38] | 11.05 | 13.25 | 15.89 | 22.88 | 18.84 | 20.21 | 19.62 | 18.63 | 17.77 | 23.94 | 23.09 | 23.28 |
| scNCL [72] | 19.89 | 20.07 | 20.05 | 49.68 | 69.50 | 73.39 | 19.93 | 19.91 | 19.71 | 45.28 | 62.55 | 62.69 |
| scBridge [36] | 26.44 | 36.34 | 42.00 | 36.01 | 39.75 | 36.15 | 28.32 | 29.69 | 32.13 | 13.13 | 26.58 | 30.08 |
| Ours | **47.43** | **52.78** | **54.95** | **52.31** | **70.18** | **77.94** | **46.40** | **48.19** | **49.92** | **51.56** | **77.37** | **78.62** |

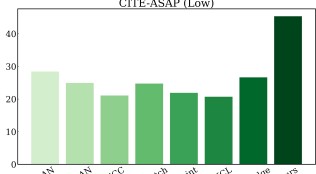
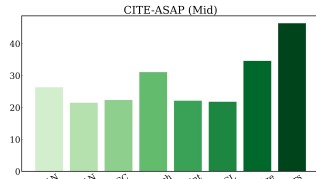
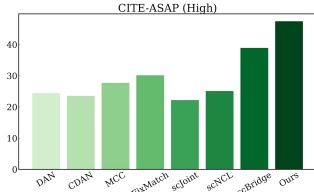

Figure 2: Qualitative comparisons (in %) on CITE-ASAP with different label ratios.

## 4.5 Summarization

In practice, we first warm up the network through labeled scRNA-seq data, and then conduct the training process in an end-to-end manner. The total loss function can be summarized as:

$$\mathcal{L} = \mathcal{L}_S + \mathcal{L}_{SL} + \lambda \mathcal{L}_{TS} + \mathcal{L}_A, \tag{16}$$

where $\lambda$ is a coefficient and set to $0.1$ empirically.

The step-by-step training algorithm of our `DANCE` is summarized in Appendix B.

## 5 Experiment

### 5.1 Experimental Settings

**Datasets.** To validate the effectiveness of the proposed `DANCE`, we conduct extensive experiments on several benchmark multi-omic single-cell datasets, with a brief introduction as follows: **Mouse Atlas Data [74].** The multi-omics data can be accessed from the Tabula Muris mouse data[3], along with the quantitative gene activity score matrix. In practice, the dataset is divided into multiple subsets based on the various sources of multi-omics data. **CITE-ASAP PBMC Data [45].** It is derived from both control and stimulated conditions. CITE-seq encompasses both antibody-derived tag (ADT) matrices and gene expression matrices, while ASAP-seq allows us to access the ADT matrices and the chromatin accessibility matrices simultaneously. In this research, we extract the gene expression matrices of CITE-seq and the chromatin accessibility matrices of ASAP-seq to construct the scRNA-seq and scATAC-seq datasets. The statistics of these datasets can be seen in Appendix D.

**Baselines.** Various approaches are adopted for performance comparisons, including three domain adaptation methods (DAN [42], CDAN [43], and MCC [26]), one semi-supervised learning method (FixMatch [52]), and three latest multi-omic single-cell data integration methods (scJoint [38], scNCL [72], and scBridge [36]). A brief introduction of these approaches is provided in Appendix E.

---

[3]https://tabula-muris.ds.czbiohub.org/

Table 2: Ablation studies (in %) of DANCE in different settings. **Bold** numbers indicate beset results.

| Dataset | snRNA_10X_v3_A-ATAC | | | snRNA_10X_v3_A-snmC | | | scRNA_10X_v2-ATAC | | | scRNA_10X_v2-snmC | | |
|---|---|---|---|---|---|---|---|---|---|---|---|---|
| Label Ratio | Low | Mid | High | Low | Mid | High | Low | Mid | High | Low | Mid | High |
| DANCE w/o OP | 31.87 | 38.55 | 40.86 | 55.60 | 61.89 | 67.69 | 33.74 | 38.31 | 44.56 | 51.66 | 59.57 | 64.18 |
| DANCE w/o SL | 48.10 | 49.09 | 49.94 | 70.04 | 76.61 | 79.00 | 49.98 | 52.56 | 54.75 | 74.08 | 79.33 | 79.89 |
| DANCE w/o TS | 48.62 | 49.01 | 49.95 | 73.85 | 77.22 | 78.91 | 49.74 | 53.07 | 55.26 | 73.99 | 79.88 | 80.85 |
| DANCE w/o CM | 44.80 | 45.63 | 45.69 | 55.91 | 59.92 | 68.79 | 46.75 | 48.01 | 49.97 | 70.81 | 76.34 | 78.02 |
| Full Model | **50.23** | **51.02** | **51.42** | **75.72** | **78.67** | **80.12** | **52.02** | **54.38** | **56.30** | **75.23** | **81.38** | **81.76** |

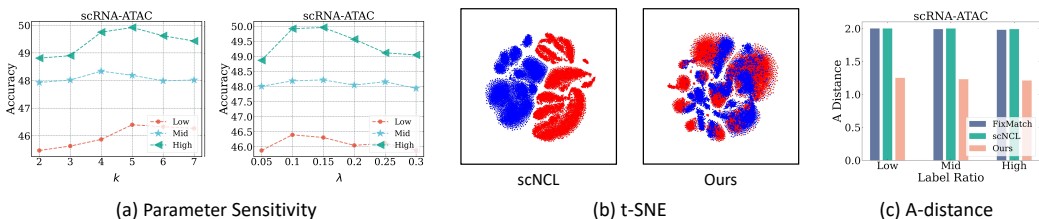

(a) Parameter Sensitivity     (b) t-SNE     (c) A-distance

Figure 3: (a) The parameter sensitivity with respect to $k$ and $\lambda$ on scRNA_SMARTer-ATAC with various label ratios. (b) The t-SNE visualization of two modalities (scRNA_10X_v2: red, ATAC: blue) with high label ratio. (c) The A-distance comparison with different label ratios.

## 5.2 Experimental Results

**Performance Comparison.** To assess the performance of various knowledge transfer methods, we conduct comprehensive quantitative experiments (Table 1) and qualitative experiments (Figure 2). From the results, several conclusions can be drawn: *Firstly*, consistent results from both quantitative and qualitative experiments reveal that DANCE significantly surpasses existing state-of-the-art approaches. We attribute this to the factor that existing methods often only address the issue of label scarcity in the target scATAC-seq data, overlooking the label scarcity issue in the source scRNA-seq data. Consequently, when cell type annotations in scRNA-seq data are limited, not only does the discriminability of representations suffer, but also the heterogeneous gap cannot be effectively eliminated. DANCE excels in three main aspects. Firstly, it utilizes OT-based pseudo-labeling to augment the scRNA-seq dataset, effectively alleviating label scarcity in scRNA-seq data. Secondly, for scATAC-seq data, we employ a divide-and-conquer strategy with consistency regularization and ambiguous set learning for two subsets of samples, addressing label scarcity in scATAC-seq data. Additionally, our proposed cross-omic multi-sample Mixup mitigates cell heterogeneity in a sample-to-sample manner. By combining these aspects, DANCE significantly outperforms previous approaches by a large margin. *Additionally*, existing methods are data-hungry, heavily relying on cell type annotations. Therefore, when the number of labels decreases, the performance drops rapidly. In contrast, DANCE is data-efficient, capable of achieving excellent performance with fewer labels. Consequently, the performance decline due to label reduction is less pronounced, thanks to our proposed dual label scarcity elimination strategy. Due to the potential class imbalance issue in single-cell data, simply increasing the number of labels may not improve performance for some methods. For example, on the snRNA_SMARTer-snmC dataset, scBridge [36] achieves a performance of 36.15% under high label ratio, which is not better than that of 39.75% under mid label ratio. In contrast, our OT-based pseudo-labeling approach considers distributing pseudo-labels in a one-to-many manner, effectively preventing biased predictions. Therefore, on all datasets, increasing the number of labels leads to corresponding performance improvements for our DANCE.

**Ablation Study.** As depicted in Table 2, we conduct comprehensive experiments on different model variants to explore the contributions of each proposed module. DANCE w/o OP represents DANCE without the proposed OT-based pseudo-labeling. DANCE w/o SL denotes DANCE without the consistency regularization for source-like samples. DANCE w/o TS indicates DANCE without the ambiguous set learning for target-specific samples. DANCE w/o CM signifies DANCE without the cross-omic multi-sample Mixup. From the results, it can be observed that the performance of DANCE w/o OP significantly decreases. This indicates that OT-based pseudo-labeling effectively alleviates the label scarcity issue in scRNA-seq data by generating pseudo-labels for source scRNA-seq data and augmenting the scRNA-seq dataset. Additionally, the slight decrease in performance of DANCE w/o SL suggests that for heterogeneous scATAC-seq data, applying consistency regularization to source-like

samples can further enhance model performance. Similarly, the minor decrease in performance of `DANCE` w/o TS suggests that target-specific samples should not be simply discarded as in previous methods. For these samples, collecting potentially correct cell types and removing them can provide weak supervision, thereby aiding performance improvement. Furthermore, the significant decrease in performance of `DANCE` w/o CM underscores the severity of the cell heterogeneity issue. Our proposed cross-omic multi-sample Mixup effectively eliminates the heterogeneous gap by generating mixed source scRNA-seq samples that simulate the distribution of target scATAC-seq data and aligning the two modalities. Thus, the effectiveness of all proposed modules is sufficiently validated.

**Sensitivity Analysis.** In Figure 3 (a), we analyze the sensitivity of two crucial hyper-parameters $k$ and $\lambda$ on scRNA_SMARTer-ATAC. Firstly, we fix the other parameters to investigate the sensitivity of the model to different values of $k$. Here, $k$ denotes the size of the partial label set and is used to control the number of potential correct cell types. As $k$ varies within the range $\{2, 3, 4, 5, 6, 7\}$, the model's performance initially increases before gradually stabilizing. Considering all three scenarios, the optimal performance is achieved at $k = 4$ or $k = 5$. Next, with other parameters fixed, we analyze the sensitivity of the coefficient $\lambda$ of $\mathcal{L}_{TS}$ in Eqn. 10. Since $\mathcal{L}_{TS}$ provides additional weak supervision by removing incorrect cell types, it typically requires a small coefficient. We change the value of $\lambda$ within the range $\{0.05, 0.1, 0.15, 0.2, 0.25, 0.3\}$. Similarly, the model's performance firstly increases and then decreases, with better performance observed at $\lambda = 0.1$.

**t-SNE Visualization.** In Figure 3 (b), we make the t-SNE [57] visualization to assess the heterogeneous gap of representations learned by scNCL [72] and our `DANCE` on scRNA_10X_v2-ATAC. Note that the higher overlap degree between the red and blue parts reflects the lower cell heterogeneity between scRNA-seq and scATAC-seq data. It can be observed that there is almost no overlap between the representations of scNCL [72], indicating that it does not address the issue of cell heterogeneity. In contrast, the representations of our `DANCE` exhibit a high degree of overlap, indicating that our approach effectively mitigates cell heterogeneity even in the presence of label scarcity.

**A-distance Comparison.** In Figure 3 (c), we compare the A-distance on scRNA_10X_v2-ATAC with various label ratios. The A-distance is calculated by employing a model and testing its ability to distinguish between the two modalities. Therefore, a low A-distance indicates a small heterogeneous gap. The A-distance is defined as: $dist_{\mathcal{A}} = 2(1 - 2\epsilon)$, where $\epsilon$ is the test error. When $\epsilon = 0$, the upper bound of $dist_{\mathcal{A}}$ is 2. This indicates that there is a significant heterogeneous gap between the two modalities, and the model can easily differentiate between them. From the results, it can be observed that the A-distance of FixMatch [52] and scNCL [72] consistently remains close to 2, indicating a significant heterogeneity between scRNA-seq data and scATAC-seq data. In contrast, `DANCE` successfully reduces cell heterogeneity, resulting in a significant decrease in the A-distance.

## 6 Conclusion

In this paper, we focus on a realistic yet underexplored problem of knowledge transfer across multi-omic single-cell data under label scarcity in both scRNA-seq and scATAC-seq data, and propose a semi-supervised framework termed `DANCE` for this problem. Specifically, `DANCE` operates dual label scarcity elimination by OT-based dataset expansion for scRNA-seq data and a divide-and-conquer strategy for scATAC-seq data. `DANCE` further introduces a sample-to-sample alignment strategy of cross-omic multi-sample Mixup to reduce cell heterogeneity. Extensive experiments on various datasets verify the superiority of `DANCE` in comparison to many state-of-the-art approaches.

**Broader Impacts and Limitations.** This study addresses the challenge of knowledge transfer across multi-omic single-cell data, which has significant potential for advancing biological research by reducing annotation costs in both source and target biological data. However, it is important to acknowledge that this work represents an initial exploration of this field and may have certain limitations. For instance, our method may not perform optimally in complex real-world scenarios, including open-set knowledge transfer and knowledge transfer under label noise. Additionally, this study primarily focuses on static scenarios, while there are complex dynamic applications, such as single-cell RNA velocity inference, that require further investigation. In future research, our goal is to address these challenges and expand our approach to encompass more generalized scenarios. To the best of our knowledge, no potential negative impacts resulting from our work have been identified.

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

# A Proof of Theorem

*Proof of Theorem 4.1.* (1) Let $\hat{P}$ be the projection of $P_n$ onto $\prod(\frac{1}{B}\mathbf{1}_B, \frac{1}{C}\mathbf{1}_C)$. Further, let $\hat{Q}$ be a solution of the Kantorovitch's problem

$$\min_{Q \in \prod(\frac{1}{B}\mathbf{1}_B, \frac{1}{C}\mathbf{1}_C)} \sum_{i,j} -Q_{ij} \log P_{ij}. \tag{17}$$

Then, by definition, we have

$$\left\langle \hat{P}, -\log P \right\rangle \geq \left\langle \hat{Q}, -\log P \right\rangle, \tag{18}$$

where $\langle \cdot, \cdot \rangle$ is the standard matrix inner product. From definition, we know that $\hat{P}$ is the solution of

$$\min_{\hat{P}} \quad \frac{1}{2} \left\| \hat{P} - P_n \right\|_F^2$$

$$\text{s.t.} \quad \hat{P}\mathbf{1}_C = \frac{1}{B}\mathbf{1}_B, \quad \hat{P}^T\mathbf{1}_B = \frac{1}{C}\mathbf{1}_C. \tag{19}$$

Using Lagrange multiplier, we know that

$$\hat{P} = P_n - \mathbf{1}_B \beta^T, \tag{20}$$

where $\beta = (\beta_1, \ldots, \beta_C)^T$ and $\beta_j = \frac{1}{B}\left(\frac{1}{B}\sum_{i=1}^B p_{ij} - \frac{1}{C}\right)$. Therefore,

$$\left\langle P_n - \hat{P}, -\log P \right\rangle = C_P. \tag{21}$$

Further, using standard OT theory, we know that as $\sigma \to \infty$, $Q$ converges to the solution of the problem in Eqn. 17 with maximal entropy, $Q_\infty$. Therefore, for large enough $\sigma$, we know that

$$\|Q - Q_\infty\|_F \leq \varepsilon / \|-\log P\|_F. \tag{22}$$

So, using Cauchy's inequality, we have

$$|\langle Q - Q_\infty, -\log P \rangle| \leq \varepsilon. \tag{23}$$

Combining Eqn. 18-23 and take $\hat{Q} = Q_\infty$ in Eqn. 18, we get

$$\langle P_n, -\log P \rangle \geq \langle Q, -\log P \rangle + C_P - \varepsilon. \tag{24}$$

Therefore, we have

$$\text{CE}(P_n, P) + \varepsilon \geq \text{CE}(Q, P) + C_P. \tag{25}$$

(2) For simplicity, we denote $\frac{L}{P} := (L_{ij}/P_{ij})_{B \times C}$. Then, we have

$$\left\langle P_n - Q, -\log \frac{L}{P} \right\rangle = \frac{1}{B}\sum P_{ij} \cdot \left(-\log \frac{L_{ij}}{P_{ij}}\right) - \sum Q_{ij} \cdot \left(-\log \frac{L_{ij}}{P_{ij}}\right)$$

$$\geq \log C_{\min} - \log C_{\max}. \tag{26}$$

Hence,

$$\langle P_n, -\log L \rangle \geq \langle Q, -\log L \rangle + C_P - \varepsilon - C_L. \tag{27}$$

$\square$

*Proof of Theorem 4.2.* From the definition, we know that

$$p_{\text{hard}} = \frac{1}{n}\sum_{i=1}^n Z_i, \ p_{\text{soft}}^t = \frac{1}{n}\sum_{i=1}^n \{tZ_i + (1-t)(1-Z_i)\}. \tag{28}$$

Therefore, $\mathbb{E}p_{\text{hard}} = \mathbb{E}Z_1 = p + \beta - 2p\beta$, $\text{Var}(p_{\text{hard}}) = \text{Var}(Z_1)/n$, $\mathbb{E}p_{\text{soft}}^t = (1-t)+(2t-1)\mathbb{E}Z_1$, and $\text{Var}(p_{\text{soft}}^t) = (2t-1)^2\text{Var}(Z_1)/n$. Hence, we have

$$\mathbb{E}\left\|p_{\text{hard}} - p\right\|^2 = (\mathbb{E}Z_1 - p)^2 + \frac{\text{Var}(Z_1)}{n},$$

$$\mathbb{E}\left\|p_{\text{soft}}^t - p\right\|^2 = \{(1-t)+(2t-1)\mathbb{E}Z_1 - p\}^2 + \frac{(2t-1)^2\text{Var}(Z_1)}{n}. \tag{29}$$

Define
$$f(t) = \mathbb{E} \left\| p_{\text{hard}} - p \right\|^2 - \mathbb{E} \left\| p_{\text{soft}}^t - p \right\|^2. \tag{30}$$

Then, from definition, we know that $f(1) = 0$. Further, we know

$$f'(t) = -2 \left\{ (1-t) + (2t-1)\mathbb{E}Z_1 - p \right\} \cdot (2\mathbb{E}Z_1 - 1) + (4-8t)\frac{\text{Var}(Z_1)}{n}$$

$$= -\left\{ 2(1 - 2\mathbb{E}Z_1)^2 + \frac{8\text{Var}(Z_1)}{n} \right\} t + \left\{ \frac{4\text{Var}(Z_1)}{n} + 2(1 - 2\mathbb{E}Z_1)(1 - \mathbb{E}Z_1 - p) \right\}. \tag{31}$$

Since $2(1 - 2\mathbb{E}Z_1)^2 + \frac{8\text{Var}(Z_1)}{n} > 0$, we know that there exists

$$t_0 = \frac{\frac{4\text{Var}(Z_1)}{n} + 2(1 - 2\mathbb{E}Z_1)(1 - \mathbb{E}Z_1 - p)}{2(1 - 2\mathbb{E}Z_1)^2 + \frac{8\text{Var}(Z_1)}{n}}$$

$$= \frac{\frac{4\text{Var}(Z_1)}{n} + 2(1 - 2p)^2(1 + 2\beta)(1 - \beta)}{\frac{8\text{Var}(Z_1)}{n} + 2(1 - 2p)^2(1 + 2\beta)^2} \in (0, 1), \tag{32}$$

such that $f'(t) < 0$ when $t \in (c_0, 1)$. So, $f(t) \geq f(1) = 0$ for $t \in (t_0, 1)$. Hence,

$$\mathbb{E} \left\| p_{\text{soft}}^t - p \right\|^2 \leq \mathbb{E} \left\| p_{\text{hard}} - p \right\|^2. \tag{33}$$

$\square$

## B  Algorithms

The step-by-step training algorithm of our DANCE is summarized in Algorithm 1. The model can get extra supervision from target-specific scATAC-seq data by removing the incorrect types from the candidate label set, the algorithm is provided in Algorithm 2.

---

**Algorithm 1** Training Algorithm of DANCE

---

**Require:** The training datasets $\mathcal{D}^{(s),l}$, $\mathcal{D}^{(s),u}$ and $\mathcal{D}^{(t)}$.
**Ensure:** The network parameters in $H(F(\cdot))$;
 1: Warm up the network using $\mathcal{D}^{(s),l}$;
 2: **repeat**
 3:   Update the pseudo-labels using Eqn. 3 and Eqn. 4;
 4:   Expanding $\mathcal{D}^{(s)}$ using Eqn. 5;
 5:   **for** $t = 1, 2, \cdots, T$ **do**
 6:     Sample a mini-batch from the training datasets;
 7:     Generate the predictions for both scRNA-seq data and scATAC-seq data by propagating the network;
 8:     Calculate the final loss using Eqn. 16;
 9:     Update the parameters of the network using backpropagation;
10:   **end for**
11: **until** convergence

---

## C  Implementation Details

All the baselines are re-implemented on NVIDIA Tesla A100 40G GPUs using PyTorch according to the original settings in the corresponding papers to ensure a fair comparison. A two-layer MLP with an embedding dimension of $64$ is employed as the encoder, and another two-layer MLP is adopted as the classifier. For all the baselines and our DANCE, we first warm up the model with labeled scRNA-seq data for 30 epochs and then train the model for another 30 epochs with a batch size of 32. Three settings with different label ratios (low: $1\%$, mid: $5\%$, high: $10\%$) are set up for experiments on each dataset to validate the sensitivity of these methods to the number of labels. We opt for SGD as the default optimizer with a learning rate of $3e-3$ and a weight decay of $1e-3$.

**Algorithm 2** Algorithm of Learning from Target-specific scATAC-seq Data

---

**Require:** Target-specific scATAC-seq dataset $\mathcal{D}^{TS}$; Threshold $\mu$; The number of classes $k$.
**Ensure:** The network parameters in $H(F(\cdot))$;
   **repeat**
      **for** $t = 1, 2, \cdots, T$ **do**
         Sample a mini-batch of $B^{TS}$ samples from $\mathcal{D}^{TS}$;
         Generate the softmax predictions for scRNA-seq data by propagating the network;
         Select the top-$k$ classes from the predictions to form $B^{TC}$ label sets;
         **for** $i = 1, 2, \cdots, B^{TC}$ **do**
            **for** $y_i^j \in Y_i$ **do**
               **if** Eqn. 8 is satisfied **then**
                  Removing cell type $y_i^j$ from $Y_i$;
               **end if**
            **end for**
         **end for**
         Providing extra guidance by loss Eqn. 10;
         Decrease the threshold $\mu$;
      **end for**
   **until** convergence

---

# D   Introduction of Datasets

We summarize the dataset information included in experiments in Table 3:

Table 3: The statistics of the multi-omic single-cell datasets used to conduct experiments.

| Dataset | Source Data Type | Source Data Size | Target Data Type | Target Data Size | Number of Cell Types | Sequence Length |
|---|---|---|---|---|---|---|
| | snRNA_10X_v3_A | 4017 | ATAC | 7962 | 18 | 18603 |
| | snRNA_10X_v2 | 7652 | ATAC | 7962 | 18 | 18603 |
| | snRNA_SMARTer | 6171 | ATAC | 7962 | 18 | 18603 |
| | scRNA_10X_v2 | 12264 | ATAC | 7962 | 18 | 18603 |
| | scRNA_SMARTer | 6288 | ATAC | 7962 | 17 | 18603 |
| Mouse Atlas Data | snRNA_10X_v3_A | 4017 | snmC | 9633 | 18 | 18603 |
| | snRNA_10X_v2 | 7652 | snmC | 9633 | 18 | 18603 |
| | snRNA_SMARTer | 6171 | snmC | 9633 | 18 | 18603 |
| | scRNA_10X_v2 | 12264 | snmC | 9633 | 18 | 18603 |
| | scRNA_SMARTer | 6288 | snmC | 9633 | 17 | 18603 |
| CITE-ASAP PBMC Data | scRNA part in CITE | 4644 | scATAC part in ASAP | 4502 | 7 | 17441 |

# E   Introduction of Baselines

Here we provide a brief introduction of the compared baseline methods as follows:

- **DAN** [42] is a deep neural network (DNN) based method for domain adaptation, which adapts the layers and the corresponding task-specific features in a layer-wise manner.

- **CDAN** [43] incorporates the discriminative information of the predictions from the classifier with adversarial learning for domain adaptation.

- **MCC** [26] is a strong domain adaptation method that achieves fast convergence speed without the explicit use of domain alignment.

- **FixMatch** [52] is a strong semi-supervised learning approach that hybrids pseudo-labeling and consistency regularization for self-training.

- **scJoint** [38] is a multi-omic single-cell data integration method based on transfer learning and semi-supervised learning, which effectively transfers labels through kNN on joint embedding space.

- **scNCL** [72] is a transfer learning framework that preserves the intrinsic structure of scATAC-seq data and employs contrastive learning to facilitate the transfer of cell types.

- **scBridge** [36] integrates multi-omic single-cell data in a heterogeneous manner by mining reliable cross-modal samples and expanding the single-cell dataset.

# F  Further Discussion

## F.1  Comparison with Classic Pseudo-labeling

As depicted in Figure 4, classic pseudo-labeling [52] assigns the category label with the highest confidence as the pseudo-label, establishing a one-to-one correspondence between a sample and a category label. This approach overlooks the potential connections between the sample distribution and the class distribution, especially in cases of class imbalance. In contrast, our proposed OT-based pseudo-labeling aligns multiple samples with a category center through the coupling matrix $Q$ and the cost matrix $-log P$. This approach considers the connections between the sample and class distribution, effectively addressing the issue of potential biased predictions in scRNA-seq data.

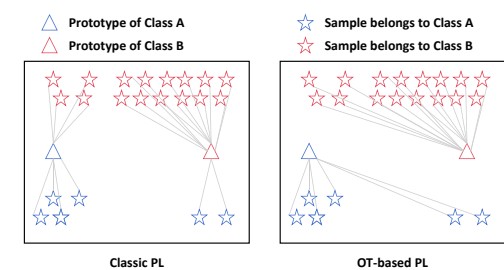

Figure 4: An illustration of classic pseudo-labeling and our OT-based pseudo-labeling.

## F.2  Advantages of Divide and Conquer for Target scATAC-seq Data

Different from traditional approaches [52, 78] that directly discard the target-specific samples, we utilize these samples to provide weak supervision for the model (algorithm can be found in Algorithm 2). By progressively removing the incorrect cell types, the model gets extra guidance from these samples. Incorporating consistency regularization for source-like samples (Figure 5) and incorrect cell types removal for target-specific samples, the entire scATAC-seq dataset $\mathcal{D}^{(t)} = \mathcal{D}^{SL} \cup \mathcal{D}^{TC}$ is fully explored, contributing to the semi-supervised knowledge transfer process.

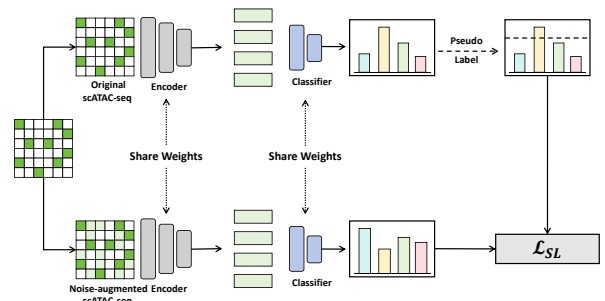

Figure 5: An illustration of consistency regularization for source-like scATAC-seq samples.

## F.3  Advantages of Cross-omic Multi-sample Mixup

Different from previous alignment strategies that rely on labels from scATAC-seq data or paired labels, our proposed cross-omic multi-sample Mixup reconstructs class-wise similarity through the weights of classifiers (Figure 6). It then fuses scRNA-seq samples by incorporating semantic information from scATAC-seq data. Without the need for labels from scATAC-seq data or pairwise labels, we transform the distribution of scRNA-seq samples to align with scATAC-seq data. This makes our alignment strategy more flexible and convenient. Considering the high cost of annotating single-cell data, our alignment strategy is more suitable for the problem of semi-supervised knowledge transfer across multi-omic single-cell data.

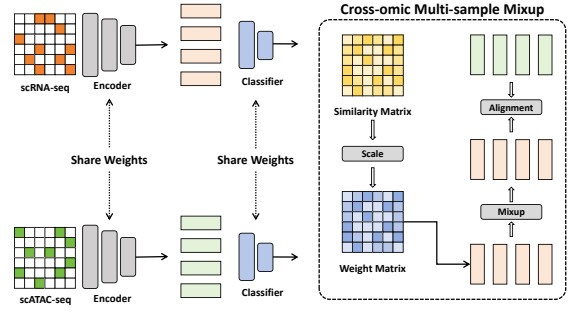

Figure 6: An illustration of cross-omic multi-sample Mixup.

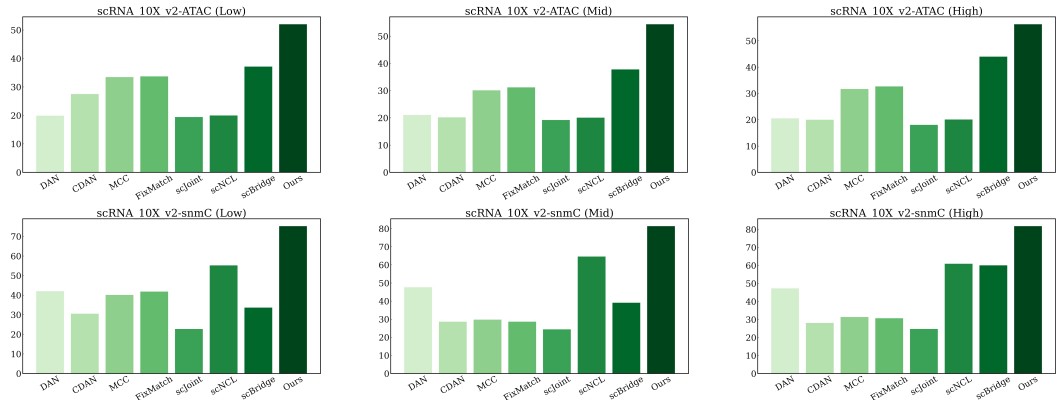

Figure 7: Qualitative comparisons (in %) with state-of-the-art approaches on additional datasets with different label ratios.

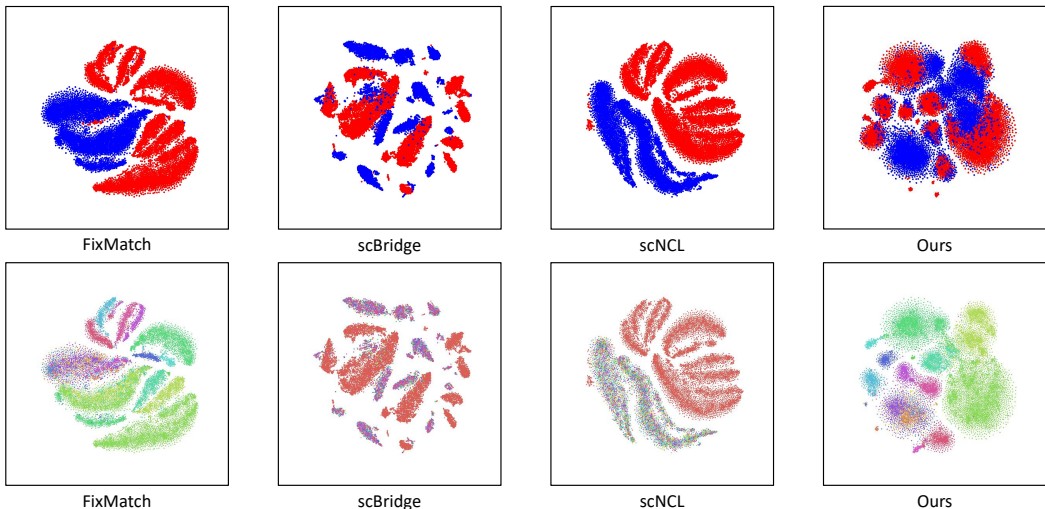

Figure 8: The t-SNE visualization of two modalities (scRNA_10X_v2: red, ATAC: blue) (the first row) and different cell types (the second row) with high label ratio.

### F.4 Further Sensitivity Analysis

In addition to these parameters, we also investigate the two empirical hyper-parameters: $\sigma$ and $\tau$. From the results in Table 4 and Table 5, we can observe that the performance is not sensitive to the choice of $\sigma$ when we change $\sigma$ from 5 to 20, so we empirically set its value to 10. We also vary the value of $\tau$ with the range of $\{0.8, 0.85, 0.9, 0.95\}$. The results below indicate that the method is not sensitive to $\tau$ in the interval $[0.8, 0.95]$. Therefore, we set $\tau$ to 0.9 as the default.

Table 4: Sensitivity analysis of $\sigma$.

| Dataset | 5 | 10 | 15 | 20 |
|---|---|---|---|---|
| CITE-ASAP | 47.01 | 47.45 | 47.32 | 47.14 |
| snRNA_10X_v3_A-ATAC | 51.16 | 51.42 | 51.08 | 51.17 |

We progressively remove the incorrect cell types for additional supervision. However, how to define an incorrect cell type may be a point of interest. To answer this question, We have included a sensitivity analysis of the threshold $\mu$ by varying it in $\{0.0005, 0.001, 0.0015, 0.002\}$. The results in Table 6 indicate that a threshold of $1e - 3$ brings good performance with significant differences. In particular, after the softmax operation, the sum of the confidences for all cell types equals 1 with

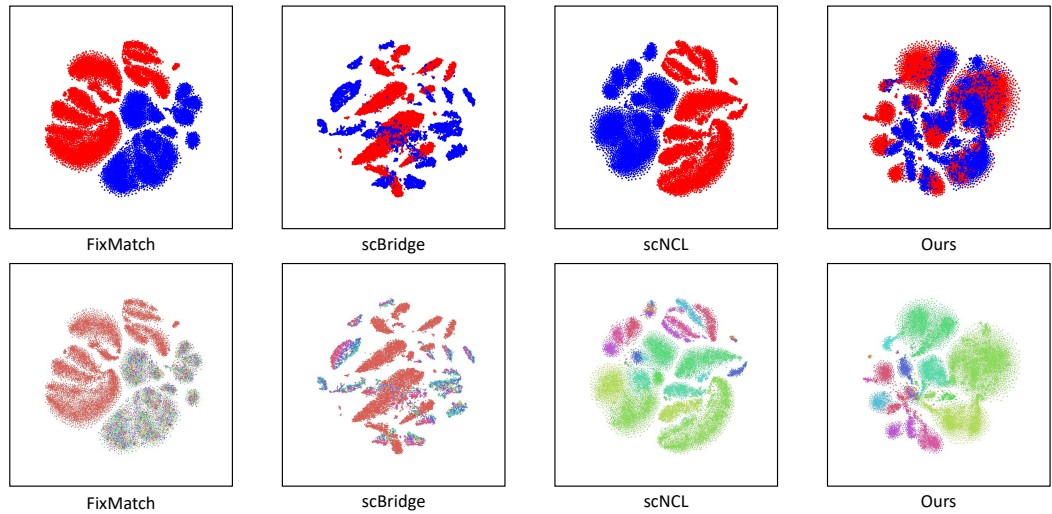

Figure 9: The t-SNE visualization of two modalities (scRNA_10X_v2: red, snmC: blue) (the first row) and different cell types (the second row) with high label ratio.

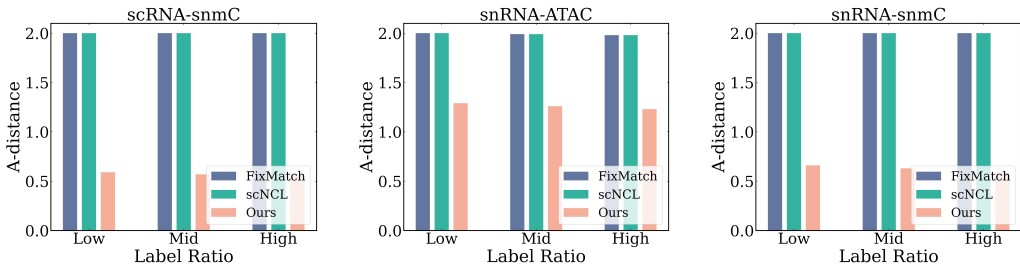

Figure 10: Comparisons of the A-distance between source scRNA-seq data and target scATAC-seq data with various label ratios.

more than $10$ cell types. In this context, the confidence scores are quite small and a difference of $1e-3$ indicates a sufficient difference.

### F.5 Further Comparison of the A-distance

Besides scRNA_10X_v2-ATAC, we further compare the A-distance on 3 more datasets, including scRNA_10X_v2-snmC, snRNA_10X_v2-ATAC, and snRNA_10X_v2-snmC in Figure 10. The results are consistent with the analysis in Section 5.2, which validates the robustness of our approach in reducing cell heterogeneity.

### F.6 Additional Qualitative Results

We showcase additional qualitative experimental results on two more datasets in Figure 7. It can still be observed that our DANCE achieves the best performance. From the results, we can obtain consistent conclusions as previously mentioned in Section 5.2.

### F.7 Additional t-SNE Visualization

In Figure 8 and Figure 9, we make the t-SNE visualization comparison with more baseline methods. From the results in the first row, it can be found that there is almost no overlap degree in FixMatch [52] and scNCL [72]. There are slight overlaps in scBridge [36], but the heterogeneous gap remains large when certain parts of labels are missing. Compared with these methods, our DANCE achieves the largest overlap degree since it effectively reduces cell heterogeneity. In the second row, each different color represents a different cell type. It can be seen that, in the case of label scarcity, our DANCE still

Table 5: Sensitivity analysis of $\tau$.

| Dataset | 0.8 | 0.85 | 0.9 | 0.95 |
|---|---|---|---|---|
| CITE-ASAP | 46.98 | 47.06 | 47.45 | 47.33 |
| snRNA_10X_v3_A-ATAC | 50.54 | 51.15 | 51.42 | 50.87 |

Table 6: Sensitivity analysis of $\mu$.

| Dataset | 5e-4 | 1e-3 | 1.5e-3 | 2e-3 |
|---|---|---|---|---|
| CITE-ASAP | 46.58 | 47.45 | 47.40 | 47.02 |
| snRNA_10X_v3_A-ATAC | 50.07 | 51.42 | 51.14 | 51.03 |

maintains good cell discriminability compared to the other three methods. This indicates that `DANCE` effectively transfers cell type knowledge from scRNA-seq data to scATAC-seq data.

Table 7: Ablation on OT strategy for target single-cell data.

| Label Ratio | Low | Mid | High |
|---|---|---|---|
| w/ OT for scATAC-seq | 34.90 | 36.87 | 39.34 |
| w/o OT for scATAC-seq (Ours) | **45.36** | **46.38** | **47.45** |

## F.8 Optimal Transport Strategy for Target Single-cell Data

Despite the effectiveness of OT-based dataset expansion for source scRNA-seq data, due to the presence of cellular heterogeneity, directly applying this strategy to target single-cell data may not be effective. We have added a model variant w/ OT for scATAC-seq to support our point. The compared results on CITE-ASAP dataset are shown in Table 7. We can find that our full model outperforms w/ OT for scATAC-seq, which validates the OT strategy may not be the optimal choice given the significant cellular heterogeneity inherent in scATAC-seq data. This is why we introduce the divide-and-conquer strategy for target samples.

Table 8: Comparison of pseudo-labeling accuracy (%).

| Label Ratio | Low | Mid | High |
|---|---|---|---|
| unlabeled scRNA-seq data | 78.85 | 79.17 | 80.13 |
| source-like scATAC-seq data | 77.12 | 78.04 | 78.56 |

## F.9 Effectiveness of Divide-and-conquer Strategy

An interesting question is whether the initial prediction's reliance on the divide-and-conquer strategy has been accurately labeled. To explore this issue, we have included a comparison of pseudo-labeling accuracy between unlabeled scRNA-seq data and source-like scATAC-seq data in Table 8. The results for the snRNA_SMARTer-snmC dataset are presented below. From the findings, it is evident that the accuracy of the source-like scATAC-seq data closely aligns with that of unlabeled scRNA-seq data, confirming the effectiveness of our approach.

Table 9: Performance comparison when transferring cell types in the opposite direction.

| Label Ratio | Low | Mid | High |
|---|---|---|---|
| scJoint | 28.14 | 45.40 | 49.59 |
| scBridge | 32.24 | 51.55 | 53.62 |
| scNCL | 36.46 | 51.01 | 52.73 |
| Ours | **70.33** | **70.78** | **72.89** |

### F.10 Cell Type Transfer in the Opposite Direction

Even though mainstream algorithms focus on transferring cell type knowledge from scRNA-seq to scATAC-seq data, we are interested in exploring the possibility of cell type transfer in the opposite direction, namely from scATAC-seq to scRNA-seq. We have conducted an experiment that performs this reverse cell type transfer on the CITE-ASAP dataset. The results presented in Table 9 confirm the effectiveness of our method in this reverse direction.

Table 10: Efficiency Comparison.

| Method | Memory Cost | Training Time/Epoch | Acc (%) |
|--------|-------------|---------------------|---------|
| scJoint | 0.7GB | 30s | 22.07 |
| scBridge | 1.4GB | 4s | 23.38 |
| scNCL | 1.0GB | 70s | 22.53 |
| Ours | 1.4GB | 30s | 46.40 |

### F.11 Computation Cost

The comparison of computation costs is also a point of interest. We have included a comparison of memory and time in Table 10, and it is evident that our method offers a competitive computation cost. Specifically, the performance of scNCL is significantly inferior to ours (our performance improvement exceeds 105.9%), while our memory cost sees only a slight increase with even less training time.

Table 11: Performance comparison with two additional domain adaptation methods.

| Dataset | CI-AS | CI-AS | CI-AS | sn-AT | sn-AT | sn-AT |
|---------|-------|-------|-------|-------|-------|-------|
| Label Ratio | Low | Mid | High | Low | Mid | High |
| SLA | 32.05 | 36.99 | 41.74 | 35.36 | 37.48 | 42.41 |
| COT | 31.44 | 35.86 | 39.01 | 35.73 | 36.79 | 43.85 |
| Seurat | 38.71 | 41.76 | 43.25 | 33.46 | 39.78 | 45.09 |
| Harmony | 39.03 | 39.99 | 44.06 | 32.17 | 39.94 | 44.56 |
| Ours | **45.36** | **46.38** | **47.45** | **50.23** | **51.02** | **51.42** |

### F.12 Further Comparison with More Baselines

To benchmark our DANCE against more baseline methods, we have included SLA [76], COT [41], Seurat [54], and Harmony [30] on different datasets for performance comparison. The results shown in Table 11 demonstrate the effectiveness of the proposed DANCE.

Table 12: Performance on full MouseAtlas data.

| Label Ratio | Low | Mid | High |
|-------------|-----|-----|------|
| scJoint | 62.36 | 67.75 | 70.02 |
| scBridge | 64.99 | 69.86 | 71.44 |
| scNCL | 61.50 | 69.04 | 71.17 |
| Ours | **70.64** | **72.21** | **72.58** |

### F.13 Performance on Full MouseAtlas Data

To evaluate the performance under different batches, we have included the experimental results on the full MouseAtlas dataset. Specifically, we collect different batches of the MouseAtlas dataset and remove some of the batches. The results in Table 12 indicate that our approach still outperforms other methods, showing the potential of DANCE in batch effect correction.

Table 13: Performance on real-world scenarios.

| Labeled scRNA-seq Data | CITE | CITE | snRNA_10X_v3_A | snRNA_SMARTer |
|---|---|---|---|---|
| Unlabeled scRNA-seq Data | snRNA_10X_v3_A | snRNA_SMARTer | CITE | CITE |
| scATAC-seq Data | ASAP | ASAP | ASAP | ASAP |
| scJoint | 86.59 | 87.27 | 75.94 | 76.53 |
| scBridge | 88.01 | 89.24 | 78.99 | 77.60 |
| scNCL | 86.24 | 88.28 | 75.80 | 75.42 |
| Ours | **93.37** | **92.48** | **84.88** | **83.79** |

## F.14 Real-world Cases

In real-world scenarios, the proposed DANCE can transfer cell types from multiple-source scRNA-seq data to target scATAC-seq data. We have included experiments to transfer cell type knowledge from a labeled scRNA-seq dataset and an unlabeled scRNA-seq dataset to scATAC-seq data. The results in Table 13 indicate that our approach is superior to other methods in practical scenarios.

