# OpenReview forum: "Semi-supervised Knowledge Transfer Across Multi-omic Single-cell Data"
_NeurIPS.cc/2024/Conference — NeurIPS 2024 poster_

### Official Review · Reviewer_L3nP · 2024-07-06

**Soundness:** 3
**Presentation:** 3
**Contribution:** 2
**Rating:** 4
**Confidence:** 4

**Summary:**

This paper introduces a novel method to address the knowledge transfer challenge in multi-omic single-cell data. Specifically, it focuses on scenarios where annotations are available partially in one modality, namely scRNA-seq data, and aims to infer annotations in another modality, the scATAC-seq data, without requiring paired datasets.

The authors proposed to use a shared encoder to project the two modalities, with additional compoenents including: optimal transport-based dataset expansion for scRNA-seq data, divide and conpuer for target scATAV-seq data, and cross-omic multi-sample mixup.

The author conducted comprehensive comparison with SOTA method and showed that the proposed method achieves the best performance.

**Strengths:**

Overall, the proposed approach is innovative, and the methodology section is well-structured. An ablation study confirms that each proposed component contributes significantly to the overall superior performance of the methology.

**Weaknesses:**

I'm not entirely convinced about the prevalence of the problem setting described by the author, where only a small subset of the scRNA-seq data is annotated. It may occur in scenarios such as large experiments conducted in batches, where only certain batches are annotated. It raises questions about the experimental setup. Are annotations removed randomly or based on specific conditions or batches? I wonder how does the availability of annotated subsets (randomly or by batch) affect the overall study results.

**Questions:**

I wonder how the methodology comparision is conducted. As the authors argued in the paper, the SOTA methods in the experimental studies does not deal with scenarios where the scRNA-seq is only partially annotated. Therefore, in the comparison experiment, such as in the "low label ratio" setup, is it that only the 1% of the scRNA-seq is used for cross-modality matching? There are popular existing methods such as Seurat and Harmony that provided a potential two-step approach: first learn the full annotation on scRNA-seq, and then transfer to scATAC-seq, which might be a more fair comparison.

**Limitations:**

The authors have addressed limitations such as only applicable to open-set settings.

---

> ### Author Rebuttal · Authors · 2024-08-07
>
> We are truly grateful for the time you have taken to review our paper and your insightful review. Here we address your concerns in the following.
>
> > Q1. I'm not entirely convinced about the prevalence of the problem setting described by the author, where only a small subset of the scRNA-seq data is annotated.
>
> A1. Thanks for your comment. The label scarcity in scRNA-seq data is practical as supported in the previous works [1,2,3]. They have developed semi-supervised approaches to address the label scarcity in the scRNA-seq data. We will include the citations in our revised version.
>
> > Q2. It may occur in scenarios such as large experiments conducted in batches, where only certain batches are annotated. It raises questions about the experimental setup. Are annotations removed randomly or based on specific conditions or batches? I wonder how does the availability of annotated subsets (randomly or by batch) affect the overall study results.
>
> A2. Thanks for your question. The experiments in the manuscript are randomly removed. We have added an experiment to compare the performance of removing annotations by batches. Specifically, we collect different batches of MouseAtlas dataset and remove some of the batches. From the following results, we can observe that our approach still outperforms other methods, which verifies the robustness of our approach in different scenarios. We will include this in our revised version.
>
> | Label Ratio | Low   | Mid   | High  |
> | ----------- | ----- | ----- | ----- |
> | scJoint     | 62.36 | 67.75 | 70.02 |
> | scBridge    | 64.99 | 69.86 | 71.44 |
> | scNCL       | 61.50 | 69.04 | 71.17 |
> | Ours        | **70.64** | **72.21** | **72.58** |
>
>
> >Q3. I wonder how the methodology comparision is conducted. As the authors argued in the paper, the SOTA methods in the experimental studies does not deal with scenarios where the scRNA-seq is only partially annotated. Therefore, in the comparison experiment, such as in the "low label ratio" setup, is it that only the 1% of the scRNA-seq is used for cross-modality matching? There are popular existing methods such as Seurat and Harmony that provided a potential two-step approach: first learn the full annotation on scRNA-seq, and then transfer to scATAC-seq, which might be a more fair comparison.
>
> A3. Thanks for your suggestion. To solve your concern, we have included Seurat [4] and Harmony [5] for performance comparison. For these two methods, we first learn learn the full annotation on scRNA-seq data, and then transfer the knowledge to scATAC-seq data. From the following results on CITE-ASAP and snRNA_10X_v3_A-ATAC, we can find that our method surpasses other methods consistently. We will include this in our revised version.
>
> | Dataset     | CI-AS | CI-AS | CI-AS | sn-AT | sn-AT | sn-AT |
> | ----------- | ----- | ----- | ----- | ----- | ----- | ----- |
> | Label Ratio | Low   | Mid   | High  | Low   | Mid   | High  |
> | Seurat      | 38.71 | 41.76 | 43.25 | 33.46 | 39.78 | 45.09 |
> | Harmony     | 39.03 | 39.99 | 44.06 | 32.17 | 39.94 | 44.56 |
> | Ours        | **45.36** | **46.38** | **47.45** | **50.23** | **51.02** | **51.42** |
>
>
>
> **Reference**
>
> [1] Dong et al., Semi-Supervised Deep Learning for Cell Type Identification From Single-Cell Transcriptomic Data, IEEE/ACM Transactions on Computational Biology and Bioinformatics, 2022
>
> [2] Kimmel et al., scNym: Semi-supervised adversarial neural networks for single cell classification, bioRxiv, 2020
>
> [3] We et al., CALLR: a semi-supervised cell-type annotation method for single-cell RNA sequencing data, Bioinformatics, 2021
>
> [4] Stuart et al., Comprehensive integration of single-cell data, Cell, 2019
>
> [5] Korsunsky et al., Fast, sensitive and accurate integration of single-cell data with Harmony, Nature Methods, 2019
>
> In light of these responses, we hope we have addressed your concerns, and hope you will consider raising your score. If there are any additional notable points of concern that we have not yet addressed, please do not hesitate to share them, and we will promptly attend to those points.

---

### Official Review · Reviewer_puzU · 2024-07-06

**Soundness:** 3
**Presentation:** 3
**Contribution:** 3
**Rating:** 7
**Confidence:** 5

**Summary:**

This paper proposes a label transfer method from scRNA-seq to scATAC-seq data. Based on the heterogeneity of single-cell data, this work partitions data into several groups and designs effective strategies to tackle them respectively. Experiments demonstrate the effectiveness of the proposed method.

**Strengths:**

1. The proposed method is technically sound, outperforming existing methods on two scRNA-seq and scATAC-seq integration datasets.
2. Ablation studies demonstrate the effectiveness of each component in the proposed method.
3. The paper is well written and clearly organized in general, which is easy to read and follow.

**Weaknesses:**

1. In the Introduction, the authors claim that "a small fraction of scRNA-seq data with cell types annotated" aligns more closely with practical scenarios. Do you have any evidence to support the claim? You may provide an example to show that scRNA-seq data are not processed and annotated as a whole, but only a small portion is annotated.
2. The uniform distribution assumption in the optimal transport process could be wrong, as most scRNA-seq data are unbalanced across different cell types.
3. In Eq. 8, the negative cell types are removed with a significant difference from the cell type with the highest probability. However, does a threshold of 1e-3 represent significant differences?
4. What does $S_i$ in Eq. 10 mean?
5. What is the motivation of mixing scRNA-seq data to form scATAC-seq data? Is such an operation biologically reasonable?
6. Typo: In line 253 cBridge -> scBridge
7. Currently the author split the MouseAtlas dataset into several subsets with different combinations for evaluation. I recommend the author provide the experimental results on the full dataset.

**Questions:**

Please refer to my concerns in the weaknesses section.

**Limitations:**

No significant limitations of this work are found.

---

> ### Author Rebuttal · Authors · 2024-08-07
>
> We sincerely appreciate the time you've taken to review our paper and for your insightful comments. Your positive feedback is highly encouraging for us! We'd like to address your concerns in the following response.
>
> > Q1. In the Introduction, the authors claim that "a small fraction of scRNA-seq data with cell types annotated" aligns more closely with practical scenarios. Do you have any evidence to support the claim? You may provide an example to show that scRNA-seq data are not processed and annotated as a whole, but only a small portion is annotated.
>
> A1. Thanks for your comment. The label scarcity in scRNA-seq data is practical as supported in the previous works [1,2,3]. They have developed semi-supervised approaches to address the label scarcity in the scRNA-seq data. We will include the citations in our revised version.
>
>
> > Q2. The uniform distribution assumption in the optimal transport process could be wrong, as most scRNA-seq data are unbalanced across different cell types.
>
> A2. Thanks for your comment. In real-world scenarios, we often have no prior knowledge about the distribution of specific cell types and adopting the uniform distribution assumption is a popular choice without prior information. In future work, we aim to extend our methods to scenarios with prior distribution information for imbalanced scenarios. We will include the discussion of potential future works in our revised version.
>
> > Q3. In Eq. 8, the negative cell types are removed with a significant difference from the cell type with the highest probability. However, does a threshold of 1e-3 represent significant differences?
>
> A3. Thanks for your question. We have included a sensitivity analysis of the threshold $\mu$ by varying it in {0.0005,0.001,0.0015,0.002}. The following results indicate that a threshold of 1e-3 brings good performance with significant differences. In particular, after the softmax operation, the sum of the confidences for all cell types equals 1 with more than 10 cell types. In this context, the confidence scores are quite small and a difference of 1e-3 indicates a sufficient difference.
>
> | Threshold $\mu$     | 5e-4  | 1e-3  | 1.5e-3 | 2e-3  |
> | ------------------- | ----- | ----- | ------ | ----- |
> | CITE-ASAP           | 46.58 | 47.45 | 47.40  | 47.02 |
> | snRNA_10X_v3_A-ATAC | 50.07 | 51.42 | 51.14  | 51.03 |
>
>
> > Q4. What does $S_i$ in Eq. 10 mean?
>
> A4. Thanks for your question. $S_i$ is a typo and it should be $Y_i$ in Eq. 9. We will correct it in the revised version.
>
> > Q5. What is the motivation of mixing scRNA-seq data to form scATAC-seq data? Is such an operation biologically reasonable?
>
> A5. Thanks for your question. We **do not** generate scATAC-seq data from scRNA-seq data. Instead, we generate virtual scRNA-seq data using the Mixup technique and then reduce the semantic gap between scATAC-seq data and this virtual scRNA-seq data in the embedding space. From a biological perspective, scATAC-seq data is characterized by extreme sparsity. This results in input matrices that are significantly heterogeneous compared to scRNA-seq data. To address the challenges in data integration, our motivation is to reduce the distance between scATAC-seq data and the mixed (virtual) scRNA-seq data in the embedding space.
>
> > Q6. Typo: In line 253 cBridge -> scBridge.
>
> A6. Thanks for pointing out the typo. We will correct it in the revised version.
>
> > Q7. Currently the author split the MouseAtlas dataset into several subsets with different combinations for evaluation. I recommend the author provide the experimental results on the full dataset.
>
> A7. Thanks for your suggestion. We have included the experimental results on the full dataset. The results below indicate that our approach still outperforms other methods. We will include this in our revised version.
>
> | Label Ratio | Low   | Mid   | High  |
> | ----------- | ----- | ----- | ----- |
> | scJoint     | 62.36 | 67.75 | 70.02 |
> | scBridge    | 64.99 | 69.86 | 71.44 |
> | scNCL       | 61.50 | 69.04 | 71.17 |
> | Ours        | **70.64** | **72.21** | **72.58** |
>
> **Reference**
>
> [1] Dong et al., Semi-Supervised Deep Learning for Cell Type Identification From Single-Cell Transcriptomic Data, IEEE/ACM Transactions on Computational Biology and Bioinformatics, 2022
> [2] Kimmel et al., scNym: Semi-supervised adversarial neural networks for single cell classification, bioRxiv, 2020
> [3] We et al., CALLR: a semi-supervised cell-type annotation method for single-cell RNA sequencing data, Bioinformatics, 2021
>
> Thanks again for appreciating our work and for your constructive suggestions. Please let us know if you have further questions.

---

> > ### Comment · Reviewer_puzU · 2024-08-12
> >
> > Thanks for the detailed responses which addressed most of my concerns. However, the current evidence of whether the setting aligns with practical scenarios is still weak. It would be more convincing if the authors could provide some real-world examples. Moreover, the experimental results on full datasets are more solid than those on subsets. They should be included in the main paper after the revision.

---

> > > ### Author Response · Authors · 2024-08-12
> > > **Thank you for your feedback!**
> > >
> > > Thank you for your feedback! We are pleased to address your further questions as follows:
> > >
> > > >Q1. Thanks for the detailed responses which addressed most of my concerns. However, the current evidence of whether the setting aligns with practical scenarios is still weak. It would be more convincing if the authors could provide some real-world examples.
> > >
> > > A1. Thanks for your suggestion. Here we provide some real-world examples.
> > >
> > > The annotation of single-cell data is quite challenging and time-consuming. For example, in PanglaoDB [1], the assignment of genes to cell types requires broad expertise to make mappings from gene markers to cell types. The complexity is further amplified by the exhaustive list of cell type markers spanning 11 columns, making the annotation process intricate and less efficient. Moreover, ground truth cell-type annotation often requires experimental validation techniques such as flow cytometry [2] and copy number variations (CNV) estimation method [3,4]. Therefore, our semi-supervised methods can save the experimental and labor costs when it comes to new datasets in the real world.
> > >
> > > Based on these examples, we can conclude that label scarcity in scRNA-seq data is a practical problem in real-world scenarios. We will include it in the revised version.
> > >
> > > >Q2. Moreover, the experimental results on full datasets are more solid than those on subsets. They should be included in the main paper after the revision.
> > >
> > > A2. Thanks for your suggestion. We will definitely include the experimental results in the revised version.
> > >
> > > **Reference**
> > >
> > > [1] Franzén et al., PanglaoDB: a web server for exploration of mouse and human single-cell RNA sequencing data, Database, 2019
> > > [2] Zhang et al., Regulatory T-cell depletion alters the tumor microenvironment and accelerates pancreatic carcinogenesis, Cancer Discov, 2020
> > > [3] Zhang et al., Single-cell analyses inform mechanisms of myeloid-targeted therapies in colon cancer, Cell, 2020
> > > [4] Kim et al., Single-cell rna sequencing demonstrates the molecular and cellular reprogramming of metastatic lung adenocarcinoma, Nature Communication, 2020
> > >
> > >
> > > Thank you again for your feedback and effort! We will add the rebuttal contents to the main paper in the final version following your valuable suggestions. Please let us know if you have further questions.

---

> > > > ### Comment · Reviewer_puzU · 2024-08-12
> > > >
> > > > Thanks for the further clarification. I totally understand that annotating single-cell data could be laboring. However, in the current setting, you randomly sample a subset to have labels. My concern is that, in practice, one dataset is more likely to be annotated as a whole, instead of sampling a 1% subset for annotation. Could you provide more explanation and evidence on that?

---

> > > > > ### Author Response · Authors · 2024-08-12
> > > > > **Thank you for your feedback!**
> > > > >
> > > > > Thank you for your feedback! We are pleased to address your further questions as follows:
> > > > >
> > > > >
> > > > > >Q1. Thanks for the further clarification. I totally understand that annotating single-cell data could be laboring. However, in the current setting, you randomly sample a subset to have labels. My concern is that, in practice, one dataset is more likely to be annotated as a whole, instead of sampling a 1% subset for annotation. Could you provide more explanation and evidence on that?
> > > > >
> > > > > A1. Thanks for your question.
> > > > >
> > > > > *Firstly*, we could have a completely new dataset without any annotation in the real world. In such a scenario, we need to make an effort to label the new data by ourselves and the proposed method can serve as an economic tool to save the labeling cost.
> > > > >
> > > > > *Secondly*, to ensure a fair and effective evaluation of different algorithms, we rely on fully labeled datasets and randomly sample a subset. Otherwise, we do not know the ground truth on test data for performance comparison. This evaluation strategy can be found in previous semi-supervised single-cell analysis works [1,2,3,4].
> > > > >
> > > > > **Reference**
> > > > >
> > > > >
> > > > > [1] Dong et al., scSemiAE: a deep model with semi‑supervised learning for single‑cell transcriptomics, BMC Bioinformatics, 2022
> > > > > [2] Dong et al., Semi-supervised Deep Learning for Cell Type Identification from Single-Cell Transcriptomic Data, IEEE/ACM Transactions on Computational Biology and Bioinformatics, 2022
> > > > > [3] Chen et al., Single-cell RNA-seq data semi-supervised clustering and annotation via structural regularized domain adaptation, Bioinformatics, 2021
> > > > > [4] We et al., CALLR: a semi-supervised cell-type annotation method for single-cell RNA sequencing data, Bioinformatics, 2021
> > > > >
> > > > > Thank you again for your feedback and effort! Please let us know if you have further questions.

---

> > > > > > ### Comment · Reviewer_puzU · 2024-08-12
> > > > > >
> > > > > > Thanks for the response. In that case, a more practical simulation would be, for example, labeling the scRNA-seq data from one source while leaving scRNA-seq data from other sources to be unlabeled.

---

> > > > > > > ### Author Response · Authors · 2024-08-13
> > > > > > > **Thank you for your feedback!**
> > > > > > >
> > > > > > > Thank you for your feedback! We are pleased to address your further questions as follows:
> > > > > > >
> > > > > > >
> > > > > > > >Q1. Thanks for the response. In that case, a more practical simulation would be, for example, labeling the scRNA-seq data from one source while leaving scRNA-seq data from other sources to be unlabeled.
> > > > > > >
> > > > > > > A1. Thanks for your suggestion. We have included experiments to transfer cell type knowledge from a labeled scRNA-seq dataset and an unlabeled scRNA-seq dataset to scATAC-seq data. The following results indicate that our approach is superior to other methods in practical scenarios.
> > > > > > >
> > > > > > > | Labeled scRNA-seq Data   | CITE           | CITE          | snRNA_10X_v3_A | snRNA_SMARTer |
> > > > > > > | ------------------------ | -------------- | ------------- | -------------- | ------------- |
> > > > > > > | Unlabeled scRNA-seq Data | snRNA_10X_v3_A | snRNA_SMARTer | CITE           | CITE          |
> > > > > > > | scATAC-seq Data          | ASAP           | ASAP          | ASAP           | ASAP          |
> > > > > > > | scJoint                  | 86.59          | 87.27         | 75.94          | 76.53         |
> > > > > > > | scBridge                 | 88.01          | 89.24         | 78.99          | 77.60          |
> > > > > > > | scNCL                    | 86.24          | 88.28         | 75.80          | 75.42         |
> > > > > > > | Ours                     | **93.37**      | **92.48**     | **84.88**      | **83.79**     |
> > > > > > >
> > > > > > >
> > > > > > >
> > > > > > > Thank you again for your feedback and effort! Please let us know if you have further questions.

---

> > > > > > > > ### Comment · Reviewer_puzU · 2024-08-13
> > > > > > > >
> > > > > > > > Thanks for the response. Now I am more convinced about the effectiveness of the proposed method, and touched by your effort in rebuttal. I will raise my score to accept.

---

> > > > > > > > > ### Author Response · Authors · 2024-08-13
> > > > > > > > > **Thanks for your feedback and increasing the rating!**
> > > > > > > > >
> > > > > > > > > Thank you for your feedback and increasing the rating! We are pleased to know that our responses have addressed your concerns. We will properly include all the rebuttal contents in the final version, following your valuable suggestions.

---

### Official Review · Reviewer_kGak · 2024-07-11

**Soundness:** 3
**Presentation:** 3
**Contribution:** 3
**Rating:** 5
**Confidence:** 5

**Summary:**

This paper introduces a semi-supervised knowledge transfer framework called DANCE, designed to effectively transfer cell type annotations from scRNA-seq data to unannotated scATAC-seq data under conditions of label scarcity. It is similar to the unsupervised domain adaptation task in computer vision. DANCE addresses the challenge of heterogeneous multi-omic data by generating pseudo-labels based on optimal transport, employing a divide-and-conquer strategy, and using cross-omic multi-sample Mixup to reduce cell heterogeneity. Extensive experiments demonstrate DANCE's superiority over state-of-the-art methods.

**Strengths:**

1. The paper is well-motivated and seems to be reproducible.
2. The paper is well-structured and easy to follow.
3. The theory involved in the method seems relatively solid.
4. The experiment fully proves the effectiveness and superiority of the DANCE.

**Weaknesses:**

1. The compared methods, especially the DA method, are relatively old. Comparison with newer methods (e.g., [a], [b], etc.) helps to understand the performance of the methods.
2. Repeated use of symbols may cause confusion and misunderstanding among readers. Eq. (1), (3), (16), and Theory 3.2 all contain $\lambda$. As far as I understand, the meanings of these symbols may be different.
3. The method includes too many empirical hyperparameters, and the ‘crucial’ parameters selected by parameter analysis seem not comprehensive enough. A more comprehensive description of each $\lambda$ and threshold $\tao$ will help to understand the method and promote future work.
4. The quality of images seems to be low, affecting comprehension. To be specific, the method process is not clearly and comprehensively shown in Figure 1. The text in Figure 2 is too small.

[a] Semi-Supervised Domain Adaptation with Source Label Adaptation
[b] COT: Unsupervised Domain Adaptation with Clustering and Optimal Transport

**Questions:**

Please refer to the strengths and weaknesses of the paper. In addition, in my opinion, the proposed method seems to be a patchwork of methods in domain adaptation, cross-modal alignment, etc. Maybe its technical innovation is questionable. A more targeted and detailed description is recommended.

**Limitations:**

The author has explained the limitations in Sec. 5.

---

> ### Author Rebuttal · Authors · 2024-08-07
>
> We greatly appreciate your time in reviewing our paper and your insightful comments. Your positive feedback is incredibly encouraging for us! We'd like to address your concerns in the following response.
>
> > Q1. The compared methods, especially the DA method, are relatively old. Comparison with newer methods (e.g., [1], [2], etc.) helps to understand the performance of the methods.
>
> A1. Thanks for your suggestion. We have included the recommended baselines, i.e., SLA [1] and COT [2] on different datasets for performance comparison. The results are shown below, which demonstrate the superiority of our method.
>
>
> | Dataset     | CI-AS | CI-AS | CI-AS | sn-AT | sn-AT | sn-AT |
> | ----------- | ----- | ----- | ----- | ----- | ----- | ----- |
> | Label Ratio | Low   | Mid   | High  | Low   | Mid   | High  |
> | SLA [1]     | 32.05 | 36.99 | 41.74 | 35.36 | 37.48 | 42.41 |
> | COT [2]     | 31.44 | 35.86 | 39.01 | 35.73 | 36.79 | 43.85 |
> | Ours        | **45.36** | **46.38** | **47.45** | **50.23** | **51.02** | **51.42** |
>
>
>
> > Q2. Repeated use of symbols may cause confusion and misunderstanding among readers. Eq. (1), (3), (16), and Theory 3.2 all contain $\lambda$. As far as I understand, the meanings of these symbols may be different.
>
> A2. Thanks for pointing out this typo. The term in Eq. 1 and Eq. 3 should be $\sigma$, and in Theory 3.2 should be $\beta$. We will correct all typos in the revised version.
>
> > Q3. The method includes too many empirical hyperparameters, and the ‘crucial’ parameters selected by parameter analysis seem not comprehensive enough. A more comprehensive description of each $\lambda$ and threshold $\tau$ will help to understand the method and promote future work.
>
> A3. Thanks for your suggestion. We have added the parameter sensitivity analysis by varying $\lambda$ (Eq. 1) and $\tau$ (Eq. 6). From the results below, we can observe that the performance is not sensitive to the choice of $\lambda$ when we change $\lambda$ from 5 to 20, so we empirically set its value to 10. We also vary the value of $\tau$ with the range of {0.8,0.85,0.9,0.95}. The results below indicate that the method is not sensitive to $\tau$ in the interval [0.8,0.95]. Therefore, we set $\tau$ to 0.9 as the default.
>
> | $\lambda$           | 5     | 10    | 15    | 20    |
> | ------------------- | ----- | ----- | ----- | ----- |
> | CITE-ASAP           | 47.01 | 47.45 | 47.32 | 47.14 |
> | snRNA_10X_v3_A-ATAC | 51.16 | 51.42 | 51.08 | 51.17 |
>
>
> | $\tau$              | 0.8   | 0.85  | 0.9   | 0.95  |
> | ------------------- | ----- | ----- | ----- | ----- |
> | CITE-ASAP           | 46.98 | 47.06 | 47.45 | 47.33 |
> | snRNA_10X_v3_A-ATAC | 50.54 | 51.15 | 51.42 | 50.87 |
>
>
> > Q4. The quality of images seems to be low, affecting comprehension. To be specific, the method process is not clearly and comprehensively shown in Figure 1. The text in Figure 2 is too small.
>
> A4. Thanks for your comment. We have revised the figures according to your suggestions and uploaded the pdf file in the global response section.
>
> > Q5. The proposed method seems to be a patchwork of methods in domain adaptation, cross-modal alignment, etc. Maybe its technical innovation is questionable. A more targeted and detailed description is recommended.
>
> A5. Thanks for your comment. We would like to point out that the innovation of our approach against existing methods is threefold:
> - **Underexplored Pratical Problem**. We focus on an underexplored yet practical problem of knowledge transfer across multi-omic single-cell data under label scarcity, while previous works focus on utilizing fully labeled scRNA-seq data.
> - **A Holistic Framework**. To both challenges of label scarcity and cell heterogeneity, we propose a holistic framework, which consists of OT-based dataset expansion and divide-and-conquer strategy for multi-omics semantic learning, followed by cross-omic multi-sample Mixup for semantics integration. All these designs are totally new for single-cell data integration.
> - **Theortical Analysis**. We provide a comprehensive theoretical analysis to support our designs, which makes our framework more solid.
> - **Superior Performance**. Our method achieves superior performance on benchmark datasets compared with state-of-the-art methods. In particular, the performance increasement on snRNA_10X_v3_A-ATAC is up to 96.4% compared to the best baseline.
>
> **Reference**
>
> [1] Yu et al., Semi-Supervised Domain Adaptation with Source Label Adaptation, CVPR 2023
> [2] Liu et al., COT: Unsupervised Domain Adaptation with Clustering and Optimal Transport, CVPR 2023
>
>
> Thanks again for appreciating our work and for your constructive suggestions. Please let us know if you have further questions.

---

> > ### Comment · Reviewer_kGak · 2024-08-14
> >
> > I appreciate the effort put into the rebuttal, which addressed some of my concerns. After reading the other reviews and replies, I  keep my positive rating.

---

> ### Author Response · Authors · 2024-08-14
> **Thank you for your feedback!**
>
> Thanks for your feedback! We are pleased to know that you keep the positive rating and your support. We will properly include all the rebuttal contents in the revised version, following your valuable suggestions.

---

### Official Review · Reviewer_W1Pi · 2024-07-31

**Soundness:** 3
**Presentation:** 3
**Contribution:** 3
**Rating:** 7
**Confidence:** 3

**Summary:**

This paper addresses the challenge of knowledge transfer across multi-omic single-cell data under label scarcity. The proposed semi-supervised framework, DANCE, uses optimal transport to generate pseudo-labels and a divide-and-conquer strategy for handling scATAC-seq data. The framework demonstrates superior performance on benchmark datasets, offering a practical solution to label scarcity in multi-omic data.

**Strengths:**

**Pros:**
1. The introduction of optimal transport (OT) into the single-cell domain effectively addresses label scarcity and imbalance issues, supported by ablation studies.
2. The divide-and-conquer strategy for scATAC-seq data and Mixup to alleviate cellular heterogeneity are well-handled.
3. The performance gain is impressive, showing improvements as label availability increases.
4. The paper is well-written and organized, making it enjoyable to read.

**Weaknesses:**

**Cons:**
1. The exclusion of OT in scATAC-seq data due to cellular heterogeneity needs practical examples or quantitative analysis to substantiate its significance.
2. The initial prediction's dependency in the divide-and-conquer strategy could lead to misclassification. Discussion on handling wrongly divided samples or providing statistics is needed.
3. Exploring if DANCE can be conducted in the opposite direction (scATAC-seq with OT and scRNA-seq with divide-and-conquer) would be beneficial.
4. Discussion on the complexity of OT and divide-and-conquer in terms of memory and time compared to existing studies is required.
5. Including a discussion on the use of scCLIP [1] in scenarios with scarce labels would enhance the paper.

[1] https://openreview.net/forum?id=KMtM5ZHxct

**Questions:**

See the Weaknesses above.

**Limitations:**

I do not observer any potential negative societal impact of this work.

---

> ### Author Rebuttal · Authors · 2024-08-07
>
> We are truly grateful for the time you have taken to review our paper, and your insightful comments and support. Your positive feedback is incredibly encouraging for us! In the following response, we would like to address your concerns and provide additional clarification.
>
> > Q1. The exclusion of OT in scATAC-seq data due to cellular heterogeneity needs practical examples or quantitative analysis to substantiate its significance.
>
> A1. Thanks for your suggestion. We have added a model variant w/ OT for scATAC-seq to support our point. The compared results on CITE-ASAP dataset are shown below. We can find that our full model outperforms w/ OT for scATAC-seq, which validates the OT strategy may not be the optimal choice given the significant cellular heterogeneity inherent in scATAC-seq data.
>
>
> | Label Ratio                  | Low   | Mid   | High  |
> | ---------------------------- | ----- | ----- | ----- |
> | w/ OT for scATAC-seq         | 34.90 | 36.87 | 39.34 |
> | w/o OT for scATAC-seq (Ours) | **45.36** | **46.38** | **47.45** |
>
> > Q2. The initial prediction's dependency on the divide-and-conquer strategy could lead to misclassification. Discussion on handling wrongly divided samples or providing statistics is needed.
>
> A2. Thanks for your suggestion. We have added the comparison of the pseudo-labeling accuracy between unlabeled scRNA-seq data and source-like scATAC-seq data. The results on the snRNA_SMARTer-snmC dataset are shown below. From the results, we can observe that the accuracy of source-like scATAC-seq data is close to that of unlabeled scRNA-seq data, verifying the effectiveness of our division.
>
> | Pseudo-labeling Acc (%)     | Low   | Mid   | High  |
> | --------------------------- | ----- | ----- | ----- |
> | unlabeled scRNA-seq data    | 78.85 | 79.17 | 80.13 |
> | source-like scATAC-seq data | 77.12 | 78.04 | 78.56 |
>
> We have also added a model variant w/o divide-and-conquer on snRNA_10X_v3_A-ATAC dataset for performance comparison. As evidenced by the results in the table below, the incorporation of the divide-and-conquer strategy yields positive gains.
>
>
> | Acc (%)                      | Low   | Mid   | High  |
> | ---------------------------- | ----- | ----- | ----- |
> | w/o divide-and-conquer       | 46.49 | 47.80 | 48.15 |
> | w/ divide-and-conquer (Ours) | **50.23** | **51.02** | **51.42** |
>
> > Q3. Exploring if DANCE can be conducted in the opposite direction (scATAC-seq with OT and scRNA-seq with divide-and-conquer) would be beneficial.
>
> A3. Thanks for your suggestion. We have included an experiment, which transfers cell type knowledge from scATAC-seq data to scRNA-seq data on the CITE-ASAP dataset. From the results below, we can validate the superiority of our method in the opposite direction.
>
>
> | Label Ratio | Low   | Mid   | High  |
> | ----------- | ----- | ----- | ----- |
> | scJoint     | 28.14 | 45.40 | 49.59 |
> | scBridge    | 32.24 | 51.55 | 53.62 |
> | scNCL       | 36.46 | 51.01 | 52.73 |
> | Ours        | **70.33** | **70.78** | **72.89** |
>
>
> > Q4. A discussion on the complexity of OT and divide-and-conquer in terms of memory and time compared to existing studies is required.
>
> A4. Thanks for your suggestion. We have included the comparison of memory and time as follows and we can observe that our method has a competitive computation cost. In particular, the performance of scNCL is much worse than ours (the performance increasement of ours is over 105.9%), while our memory cost only increases a little with even less training time.
>
> | CITE-ASAP | Memory Cost | Training Time/Epoch | Acc (%) |
> | --------- | ----------- | -------------------- | ------- |
> | scJoint   | 0.7GB       | 30s                  | 22.07   |
> | scBridge  | 1.4GB       | 4s                   | 23.38   |
> | scNCL     | 1.0GB       | 70s                  | 22.53   |
> | Ours      | 1.4GB       | 30s                  | **46.40**   |
>
> > Q5. Including a discussion on the use of scCLIP [1] in scenarios with scarce labels would enhance the paper.
>
> A5. Thanks for your suggestion. We will include the discussion in the revised version as follows.
>
> "*scCLIP [1] introduces a contrastive learning approach to integrate multi-omic single-cell data. It aligns the representations of pairwise multi-omic single-cell data without the usage of cell type labels. In contrast, we focus on the cell type knowledge transfer task in label scarcity conditions*."
>
>
> **Reference**
>
> [1] Xiong et al.,scCLIP: Multi-modal Single-cell Contrastive Learning Integration Pre-training, NeurIPS Workshop 2023
>
>
> Thanks again for appreciating our work and for your constructive suggestions. Please let us know if you have further questions.

---

> > ### Comment · Reviewer_W1Pi · 2024-08-14
> >
> > Thank you for all the efforts on rebuttal. My concerns have been alleviated, and I would like to increase my score to 7.

---

> > > ### Author Response · Authors · 2024-08-14
> > > **Thanks for your feedback and increasing your score!**
> > >
> > > Thanks for your feedback and increasing your score! We are pleased to know that your concerns have been alleviated. We will properly include all the rebuttal contents in the revised version, following your valuable suggestions.

---

### Author Rebuttal · Authors · 2024-08-07

Dear Reviewers,

We thank you for your careful reviews and constructive suggestions. We acknowledge the positive comments such as "effective and well-handled method" (Reviewer W1Pi), "impressive performance" (Reviewer W1Pi), “well-written and organized” (Reviewer W1Pi, Reviewer puzU), "well-motivated and reproducible" (Reviewer kGak), "well-structured and easy to follow” (Reviewer kGak), "solid theory” (Reviewer kGak), "good experiment” (Reviewer kGak), "technically sound method” (Reviewer puzU), “effective ablation studies” (Reviewer puzU, Reviewer L3nP), “innovative approach” (Reviewer L3nP).

We have also responded to your questions point by point. The revised figures are uploaded in the pdf file.

Please let us know if you have any follow-up questions. We will be happy to address them.

Best regards,

the Authors

---

### Author Response · Authors · 2024-08-14
**Summary of Rebuttal**

Dear Reviewers and Area Chairs,

We would like to express our sincere gratitude for your great efforts, insightful comments, support, and the constructive suggestions you have provided once again! Through our discussions and the reviewers' responses, it appears that we have effectively addressed the major concerns raised by reviewers, and received higher scores from Reviewer W1Pi and Reviewer puzU. This outcome has greatly benefited us, and we would like to thank all of you for your valuable support!

The reviewers held many positive comments on our paper, including "**solid theory**” (Reviewer kGak), "effective and well-handled method" (Reviewer W1Pi), "**impressive performance**" (Reviewer W1Pi), “well-written and organized” (Reviewer W1Pi, Reviewer puzU), "well-motivated and reproducible" (Reviewer kGak), "well-structured and easy to follow” (Reviewer kGak), "good experiment” (Reviewer kGak), "technically sound method” (Reviewer puzU), “**effective ablation studies**” (Reviewer puzU, Reviewer L3nP), and “**innovative approach**” (Reviewer L3nP).

The reviewers also raised insightful and constructive concerns. We made every effort to address all the concerns by providing detailed clarification and requested results. Here is the summary of the major revisions:

* We have added more **ablation studies** and **parameter analysis** to show the effectiveness and reliability of each component including OT and divide-and-conquer strategy.
* We have added **efficiency analysis** to make the paper more complete.
* We have included **more competing baselines** including SLA, COT, Seurat, and Harmoy to demonstrate the superiority of our approach.
* We have provided **more references** and **real-world examples** to clarify the practical usage of our method and the rationale of our settings.
* We have demonstrated **more performance comparisons** in different experimental settings to validate the effectiveness of the proposed method in the real world.

We firmly believe that our framework for knowledge transfer across multi-omic single-cell data plays a significant role in advancing the community. And we are committed to making our complete code and training details publicly available. All the rebuttal contents will be properly included in the final version, following your valuable suggestions. Moreover, we are eager to engage in further discussions with you to enhance our understanding of the domain and further improve the quality of the paper.

Once again, thank you for your time and effort in reviewing our work. We greatly appreciate your assistance in improving our manuscript!

Best regards,

the Authors

---

### Decision · Program_Chairs · 2024-09-25

**Decision:**

Accept (poster)

**Comment:**

This paper proposes a method to transfer knowledge from single-cell RNA sequencing (scRNA-seq) to single-cell ATAC-sequencing (scATAC-seq), a problem that can be formulated as domain adaptation. The proposed method, Dual Label Scarcity Elimination with Cross-omic Multi-sample Mixup, or DANCE for short, achieves this knowledge transfer through several steps, including expanding the labeled RNA data via pseudo-labeling and using a confidence-based divide-and-conquer algorithm to identify easy/hard scATAC-seq samples with respect to scRNA-seq, thereby efficiently transferring knowledge to the ATAC sequencing domain.

The paper initially received scores of 2x6 (weak accept), 1x5 (borderline accept), and 1x4 (borderline reject). After the rebuttal and during the author-reviewer discussion period, the scores improved to 2x7 (accept), 1x5 (borderline accept), and 1x4 (borderline reject). The reviewers acknowledged the novelty of the idea within the context of the problem, and the AC agrees with this assessment, recommending the paper for acceptance. **Congratulations**.

The AC suggests including all the results discussed during the rebuttal and the author-reviewer discussions. Additionally, the following two aspects should be addressed:

1. Provide an intuitive explanation of the theorems and their implications in the main text. Currently, the theorems appear somewhat disconnected, and an intuitive explanation will help readers appreciate them better.

2. The choice of marginals in the optimal transport (OT) implies that mini-batches should be uniform across the class distribution. This should be discussed and explained.

Congratulations again.